# Neural substrates, dynamics and thresholds of galvanic vestibular stimulation in the behaving primate

Annie Kwan[1], Patrick A. Forbes [2,3,4], Diana E. Mitchell[5], Jean-Sébastien Blouin[4] & Kathleen E. Cullen [5,6]

Galvanic vestibular stimulation (GVS) uses the external application of electrical current to selectively target the vestibular system in humans. Despite its recent popularity for the assessment/treatment of clinical conditions, exactly how this non-invasive tool activates the vestibular system remains an open question. Here we directly investigate single vestibular afferent responses to GVS applied to the mastoid processes of awake-behaving monkeys. Transmastoid GVS produces robust and parallel activation of both canal and otolith afferents. Notably, afferent activation increases with intrinsic neuronal variability resulting in constant GVS-evoked neuronal detection thresholds across all afferents. Additionally, afferent tuning differs for GVS versus natural self-motion stimulation. Using a stochastic model of repetitive activity in afferents, we largely explain the main features of GVS-evoked vestibular afferent dynamics. Taken together, our results reveal the neural substrate underlying transmastoid GVS-evoked perceptual, ocular and postural responses—information that is essential to advance GVS applicability for biomedical uses in humans.

[1] Department of Biomedical Engineering, McGill University, Montreal, QC H3G 1Y6, Canada. [2] Department of Neuroscience, Erasmus MC, University Medical Center Rotterdam, Rotterdam 3000 CA, The Netherlands. [3] Department of BioMechanical Engineering, Delft University of Technology, Delft 2628 CD, The Netherlands. [4] School of Kinesiology, University of British Columbia, Vancouver, BC V6T 1Z1, Canada. [5] Department of Physiology, McGill University, Montreal, QC H3G 1Y6, Canada. [6] Department of Biomedical Engineering, Johns Hopkins University, Baltimore 21205 MD, USA. Correspondence and requests for materials should be addressed to K.E.C. (email: kathleen.cullen@jhu.edu)

As we move through the world, the vestibular system in the inner ear detects our head motion relative to space, providing the brain with vital information needed for stabilizing gaze, and maintaining balance and posture. Investigating the vestibular system in isolation, however, is complicated by the fact that natural vestibular stimuli (i.e., motion) often activate other sensory inputs (i.e., proprioceptive, somatosensory, and visual). In this context, galvanic vestibular stimulation (GVS), in which current is applied to electrodes placed on the mastoid processes[1], has become an increasingly popular non-invasive tool to selectively probe and perturb vestibular function in humans. Specifically, transmastoid GVS is delivered to activate vestibular primary afferents, in turn evoking both ocular[2–7] and postural[8–14] responses, and producing a sensation of virtual self-motion[15–17]. GVS is also rapidly gaining popularity for the assessment and treatment of a wide range of clinical disorders including: Parkinson's disease[18], stroke[19], cerebellar ataxia[20], vestibulopathy[21], concussion[22], weight loss[23], and even potentially patients with aberrant reward sensitivity[24].

Considering the growing use of GVS in the clinic, it is essential to develop a comprehensive understanding of how the vestibular system responds to GVS. This knowledge is required to establish the physiological targets of both clinical tests and their interventions. To date, the induced behavioral responses in human GVS studies have led to conflicting interpretations whether GVS-evoked vestibular responses are predominantly driven by the activation of the semicircular canals or otoliths, or a combination of both types of sensory organs. Notably, one view is that otolith afferents have lower thresholds to GVS and so are preferentially activated by stimulation[25,26,3]. This proposal stems from GVS-evoked behaviours such as tonic ocular torsion[2,3], modulation of muscle sympathetic nerve activity[27,28], and static postural sway[8,10,29] as being consistent with activation of the otolith system. On the other hand, balance, navigation and perceptual responses evoked by GVS have also been modeled based on the assumption that all otolith and semicircular afferents are activated by the electrical current[1,30,31]. Despite many efforts in interpreting these evoked behavioral responses, the uncertainty of their afferent origin leaves open the question of how vestibular afferents respond to GVS applied to the mastoid processes. Furthermore, the animal studies used to understand the influence of electrical stimulation on vestibular afferents delivered stimulation inside the ear such that current is applied in much closer proximity to the vestibular end organs[32–36]. This has become a limitation in modeling the physiological basis of GVS-evoked behaviors in human studies, where vestibular afferents are instead activated via external transmastoid stimulation[1,30].

Accordingly, here we recorded from vestibular afferents while delivering GVS to surface electrodes placed behind the ears of macaque monkeys in a binaural bipolar configuration—a setup typically used in humans. We first developed and validated a primate-based model by establishing that GVS primarily evokes torsional eye movements comparable to those measured when GVS is applied to humans. We next recorded from individual vestibular afferents and established that both canal and otolith afferents respond robustly to GVS, showing parallel increases in response gain with increasing stimulation frequency as well as the higher sensitivity of irregular versus regular firing afferents. Notably, afferent responses became progressively more dynamic with increasing frequency, overturning the view that GVS-evoked responses are phasically invariant across stimulation frequencies. Furthermore, the observed high pass tuning of both types of afferents differed from their response to natural motion stimulation. Using a simple stochastic model of repetitive activity in vestibular afferents, we were able to explain the main trends of the

observed GVS-evoked response dynamics for both regular and irregular afferents. Finally, we report for the first time the neuronal detection thresholds of vestibular afferents to GVS. Comparison of regular and irregular afferents revealed similar thresholds across all afferent classes for both the canal and otolith systems, indicating that despite the lower sensitivity of regular afferents, they transmit equivalent levels of information to central vestibular pathways to detect GVS-evoked sensations of self-motion. Taken together, our results directly establish that transmastoid GVS evokes similar high pass tuning and neuronal detection thresholds in both semicircular canal and otolith afferents, providing key information in the development of physiologically accurate models of GVS activation of the vestibular system. Such models will be required for the future advancement and accurate application of this technique as a clinical tool in humans.

## Results

**Validation of a primate model for transmastoid GVS.** During experiments, the animal was comfortably seated on a motion platform with transmastoid GVS electrodes attached in a binaural bipolar configuration (see Methods section). We first established that GVS in our nonhuman primate model evoked behavioral responses comparable to those measured in humans. Then to understand the link between sensory activation and these responses, we recorded from individual canal and otolith afferents during GVS. Finally, we measured and established the differences in neuronal response dynamics of vestibular afferents to stimulation by GVS versus natural motion.

To first validate our nonhuman primate model, we investigated the effect of GVS on eye movements while monkeys fixated on a visual target projected on a cylindrical screen. Notably, prior studies in human subjects have shown that the application of sinusoidal GVS under similar conditions evokes eye movements that are primarily in the torsional plane[6]. Using a comparable experimental approach (Fig. 1a, see Methods section), we recorded eye movements while GVS was applied over a range of frequencies (0.5–8Hz) with current amplitude held fixed at 1 mA. Figure 1b illustrates the eye movements evoked by GVS at an example frequency (2 Hz) for each monkey. Overall, consistent with previous studies in humans, we found that GVS in monkey predominately elicited torsional eye movements.

Figure 1c plots the gain (normalized to responses evoked by 0.5 Hz GVS) and phase of the torsional eye velocity evoked by GVS in each monkey as a function of frequency. Across animals, gain remained relatively constant across frequencies (Fig. 1c) (repeated-measures ANOVA $F_{4,8} = 0.72$, $p = 0.604$), while phase decreased as a function of frequency (repeated-measures ANOVA $F_{4,8} = 59.25$, $p < 0.001$) such that torsional eye velocity increasingly lagged the stimulation waveform at higher frequencies. We also considered the effect of varying current amplitude (0.5–1.5 mA) on GVS-evoked eye movement responses, holding the GVS modulation frequency fixed at 2 Hz. Figure 1d plots the amplitude (normalized to responses evoked by 0.5 mA) and phase of the torsional eye velocity evoked by GVS in each monkey as a function of current amplitudes. The magnitude of torsional eye velocity increased as a function of stimulation amplitude (repeated-measures ANOVA $F_{4,8} = 12.17$, $p = 0.002$), while phase remained constant (repeated-measures ANOVA $F_{4,8} = 0.53$, $p = 0.715$).

Taken together, these findings are consistent with prior human GVS studies[6]. Additionally, we found that GVS consistently evoked larger eye movements in Monkey B than Monkeys H and D (insets in Fig. 1c, d). As discussed below, this difference in torsional eye movement sensitivity was linked to differences in afferent sensitivity to GVS: both afferent response and eye

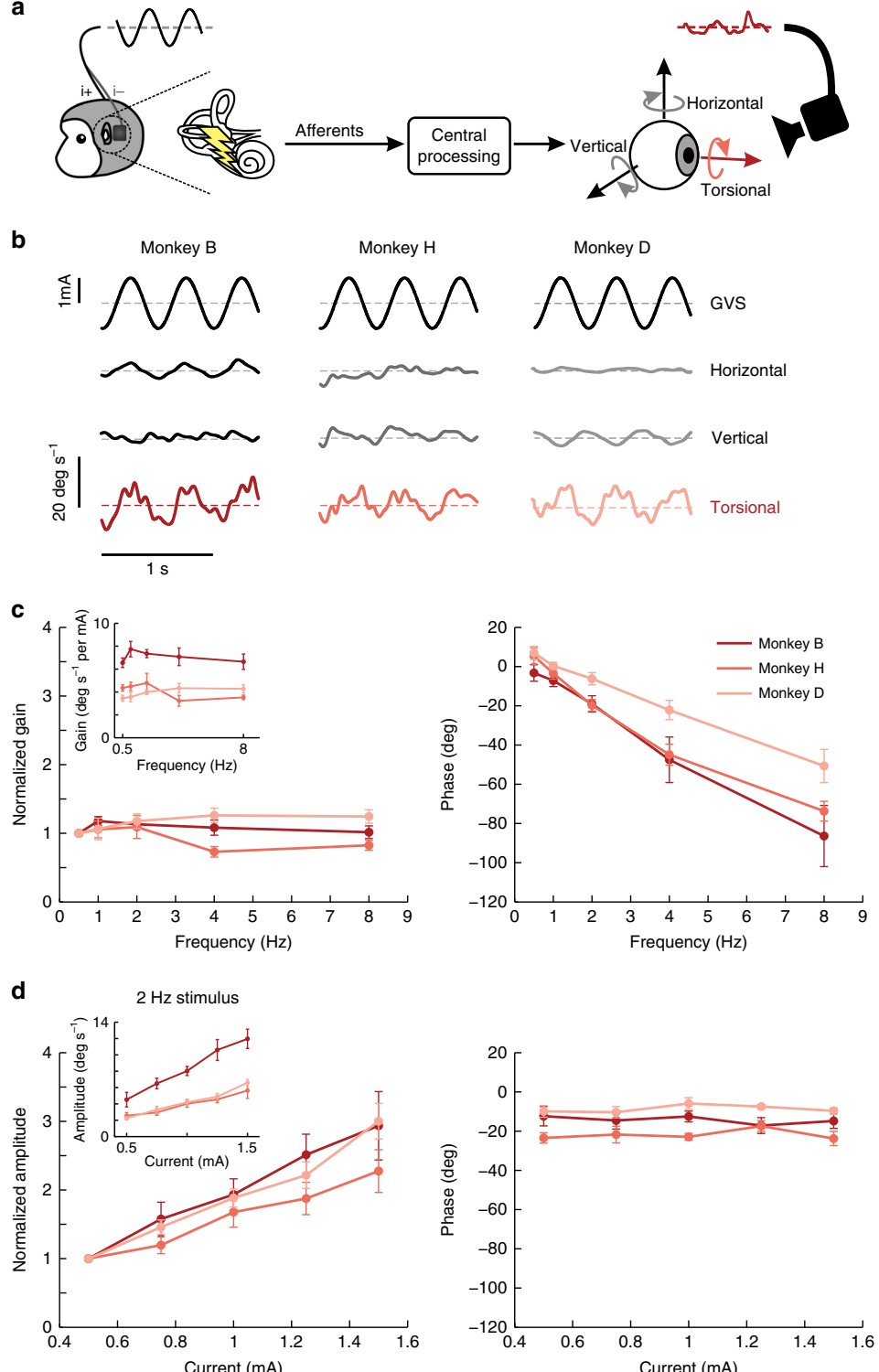

**Fig. 1** Torsional eye movements in response to sinusoidal GVS. **a** While applying sinusoidal GVS between surface electrodes placed on the mastoid processes behind the ears, we recorded the animal's eye movements while it was fixating. **b** Example traces of horizontal, vertical and torsional velocity for three animals to a 2 Hz sinusoidal stimulation of 1 mA in current amplitude. Note that the primary eye component is in the torsional plane because the animals were fixating on a target. **c** Average of the normalized gain and phase of the torsional eye velocity for each of the animals across five trials of sinusoidal stimulation that varied in frequency. The gain was normalized based on the responses at 0.5 Hz. The inset shows the gains prior to normalization. **d** Average of the normalized amplitude and phase of the torsional eye velocity for each of the animals across five trials of 2 Hz sinusoidal stimulation that varied in current amplitude. The amplitude was normalized based on the responses at 0.5 mA. The inset shows the amplitudes prior to normalization. Error bars depict the s.e.m.

movement sensitivities were larger in Monkey B than Monkey H (see Supplementary Fig. 1).

**GVS evokes robust responses in canal and otolith afferents.** To understand the neural basis of GVS activation, we next recorded from individual semicircular canal and otolith afferents ($N = 203$) in the VIIIth cranial nerve in monkeys B & H. Based on differences in their firing rates (resting discharge) in the absence of vestibular stimulation, both semicircular canal and otolith afferents can be classified as either regular or irregular (Fig. 2a; see ref. [37] for review). Our dataset consisted of $N = 119$ semicircular canal afferents, of which $N = 63$ were classified as regular (mean $CV^* = 0.06 \pm 0.00$) and $N = 56$ were classified as irregular (mean $CV^* = 0.35 \pm 0.02$). The mean resting discharge rates were $111.2 \pm 3.1$ spk s$^{-1}$ and $96.3 \pm 4.4$ spk s$^{-1}$, respectively. The remaining $N = 84$ afferents were otolith afferents, of which $N = 30$ were regular (mean $CV^* = 0.05 \pm 0.00$) and $N = 54$ were irregular (mean $CV^* = 0.38 \pm 0.02$). The mean resting discharge rates were $79.3 \pm 5.2$ spk s$^{-1}$ and $62.1 \pm 4.2$ spk s$^{-1}$, respectively.

We first recorded each afferent's response to sinusoidal GVS over a broad range of frequencies relevant to natural head rotations (0.1–25 Hz)[38]. As reviewed above, a prevailing view has been that GVS preferentially activates the otolith system[25,26], however it has been argued that activation of semicircular canals makes a significant contribution to observed behavioral responses[1,39,40]. To test between these two hypotheses, we quantified and compared the responses of canal and otolith afferents evoked by sinusoidal GVS. Figure 2b, c illustrate the responses of example semicircular canal and otolith afferents, respectively, to 1 and 8 Hz GVS. Consistent with the second hypothesis, we found that GVS evoked robust responses in semicircular canal as well as otolith afferents. In addition, both irregular canal and otolith afferents demonstrated greater modulation than their regular counterparts (Fig. 2b, c: compare bottom and top panels).

To quantify these findings, we first computed the response gain and phase at each frequency for our populations of canal and otolith afferents. Figure 2d, e plot the response gains and phases, as a function of frequency, for our canal and otolith afferents, respectively. Response gains for both regular and irregular afferents from the canals as well as otoliths increased as a function of frequency (Student $t$-test, 0.1 versus 25 Hz: for all afferent types, $p < 0.001$). Overall, irregular canal afferents (Fig. 2d) displayed significantly higher gains as a function of frequency than regular canal afferents (Student $t$-test: for all frequencies, $p < 0.0056$, Bonferroni's correction for multiple comparisons). In contrast, their response phase was comparable for both regular and irregular canal afferents (Student $t$-test: for all frequencies, $p > 0.0056$, Bonferroni's correction for multiple comparisons), while showing a marked increase as a function of frequency (Student $t$-test, 0.1 versus 25 Hz: for all afferent types, $p < 0.001$). Notably, the dynamics of these phasic responses are strikingly different than those previously reported by studies in which sinusoidal current was delivered internally into the inner ear[33,36,41]. Quantitatively similar results were obtained from our analysis of otolith afferents (Fig. 2e). Overall, irregular otolith afferents displayed significantly higher gains as a function of frequency than their regular counterparts (Student $t$-test: for all frequencies, $p < 0.0056$, Bonferroni's correction for multiple comparisons), while their response phase was comparable (Student $t$-test: for all frequencies except for 1 Hz, $p > 0.0056$, Bonferroni's correction for multiple comparisons). Further, we fit transfer functions to the population responses of both canal and otolith afferents (Fig. 2d, e; fit with a two-zero and two-pole transfer function and fractional order transfer function,

respectively), that describe the relationship between GVS and evoked afferent activity. Thus, taken together, our results show that both canal and otolith afferent populations exhibit robust responses to GVS that show increased gain and phase lead as a function of frequency.

**Afferent responses are related to firing rate regularity.** Comparison of Fig. 2d, e suggests that GVS similarly activates canal and otolith afferents, such that a given afferent's sensitivity to GVS predominately depends on its firing rate regularity (i.e., regular versus irregular) rather than the class of vestibular sensory organ that it innervates. To directly test this proposal, we plotted each afferent's response gain and phase as a function of its regularity ($CV^*$) for GVS at 1 and 8 Hz (Fig. 3a, b, respectively). Data from canal afferents (green circles) and otolith afferents (purple circles) are superimposed. At each frequency, both canal and otolith afferent response gain increased as a function of $CV^*$. These relationships were well fit by a power-law[42]. The confidence intervals of the power-law coefficients estimated for canal versus otolith afferent data overlapped, confirming that both classes of afferents showed a similar relationship between response $CV^*$ and GVS response gain. Accordingly, we estimated a single power-law fit ($a \cdot CV^{*b}$) across both afferents groups (Fig. 3a, b left panel: dotted lines). We further observed that the phase lead of both canal and otolith afferents at each frequency remained relatively constant (at ~15 and 25 degs for 1 and 8 Hz, respectively) regardless of their $CV^*$. Accordingly, we fit the relationship between phase lead and $CV^*$ with a straight line for each dataset ($a \cdot CV^* + b$), and found that indeed confidence intervals of the estimated coefficients for canal versus otolith afferents overlapped and each slope was approximately zero. Therefore, we estimated a single linear fit across both afferent groups (Fig. 3a, b right panel: dotted lines). Overall, we conclude that GVS activates both canal and otolith afferents in a similar manner that predominately depends on their firing rate regularity rather than on which class of end organ they innervate.

**Comparison of vestibular afferent response to GVS and motion.** So far we have established that canal and otolith afferents are equally well activated by transmastoid GVS; response magnitudes increase as a function of frequency and both irregular canal and otolith afferents display significantly greater modulation to GVS than do their regular counter parts. Next, we investigated how these GVS-evoked responses compare to responses evoked by natural motion stimuli. Figure 4a illustrates the responses of example regular and irregular canal afferents (top and bottom panels, respectively) for 1 Hz versus 8 Hz rotations. Consistent with previous studies[43,44], the irregular afferent was more sensitive than the regular afferent, and both afferents' sensitivities increased for the higher frequency rotation.

Figure 4b summarizes the gain and phase results for both canal afferent populations across the complete testing range (0.5–16 Hz) (dashed lines), where the increasing gain and phase lead as a function of frequency is consistent with prior reports[43,44]. To facilitate comparison of these motion-evoked responses with GVS-evoked responses, the population averaged response data quantified above in Fig. 2d were superimposed (Fig. 4b; solid lines). We observed that gain and phase responses evoked by both GVS and natural motion increased with frequency. During rotational motion, however, afferent responses exhibited greater high-pass tuning (i.e., higher gain slopes) compared to GVS when gain was normalized at 0.5 Hz. Specifically, comparison of normalized gains across matching frequencies above 0.5 Hz revealed this increase was significant for both irregular (Student $t$-test: for all frequencies, $p < 0.01$

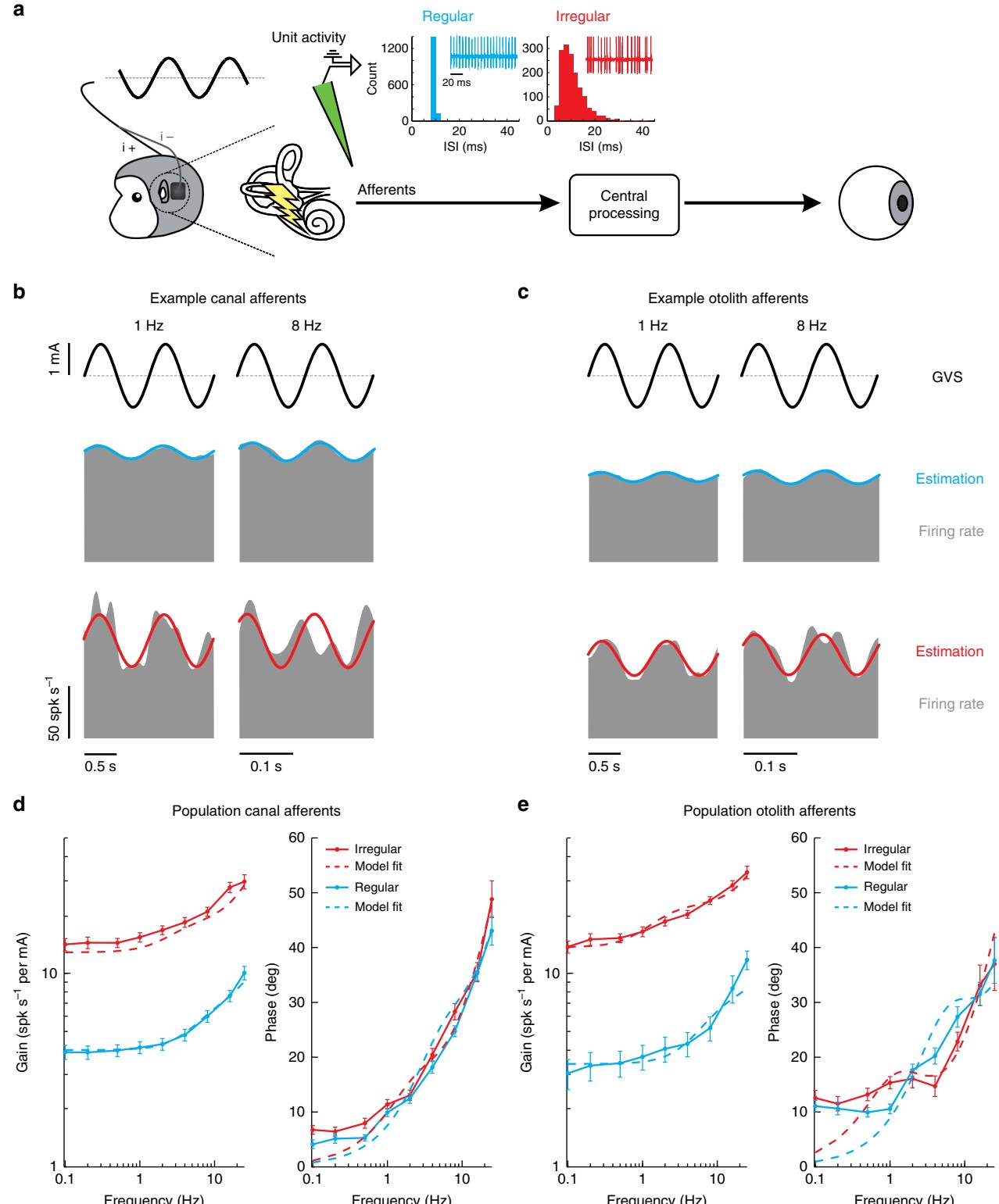

**Fig. 2** Response dynamics of vestibular afferents to sinusoidal GVS. **a** We recorded extracellular single-unit activity from vestibular afferents using tungsten electrodes during sinusoidal GVS. The insets show the interspike interval (ISI) histogram for the resting discharge of the example canal afferents. Regular afferents have a narrow ISI distribution whereas irregular afferents have a broader ISI distribution. **b** Firing rate (gray) of example regular (blue) and irregular (red) canal afferents to 1 Hz (left) and 8 Hz (right) sinusoidal GVS. Firing rate estimates (solid blue and red lines) were found using Eq. 1. **c** Firing rate (gray) of example regular (blue) and irregular (red) otolith afferents to 1 Hz (left) and 8 Hz (right) sinusoidal GVS. **d** Population averaged gain (left) and phase (right) of regular (blue) and irregular (red) canal afferents. Dashed lines depict the transfer function fits of response dynamics for regular and irregular canal afferent activity using Eq. 2. (regular: $H(s) = \frac{98(s+26)(s+188)}{(s+47)(s+2578)}$; irregular: $H(s) = \frac{418(s+12)(s+136)}{(s+18)(s+2739)}$). **e** Population averaged gain (left) and phase (right) of regular (blue) and irregular (red) otolith afferents. Dashed lines depict the transfer function fits of response dynamics for regular and irregular otolith afferent activity using Eq. 3, (regular: $H(s) = 3.15\frac{s^{0.09}(1+0.019s)^{1.30}}{(1+0.014s)}$; irregular: $H(s) = 1.38\frac{s^{0.12}(1+0.009s)^{1.46}}{(1+0.009s)}$). Error bars depict the s.e.m.

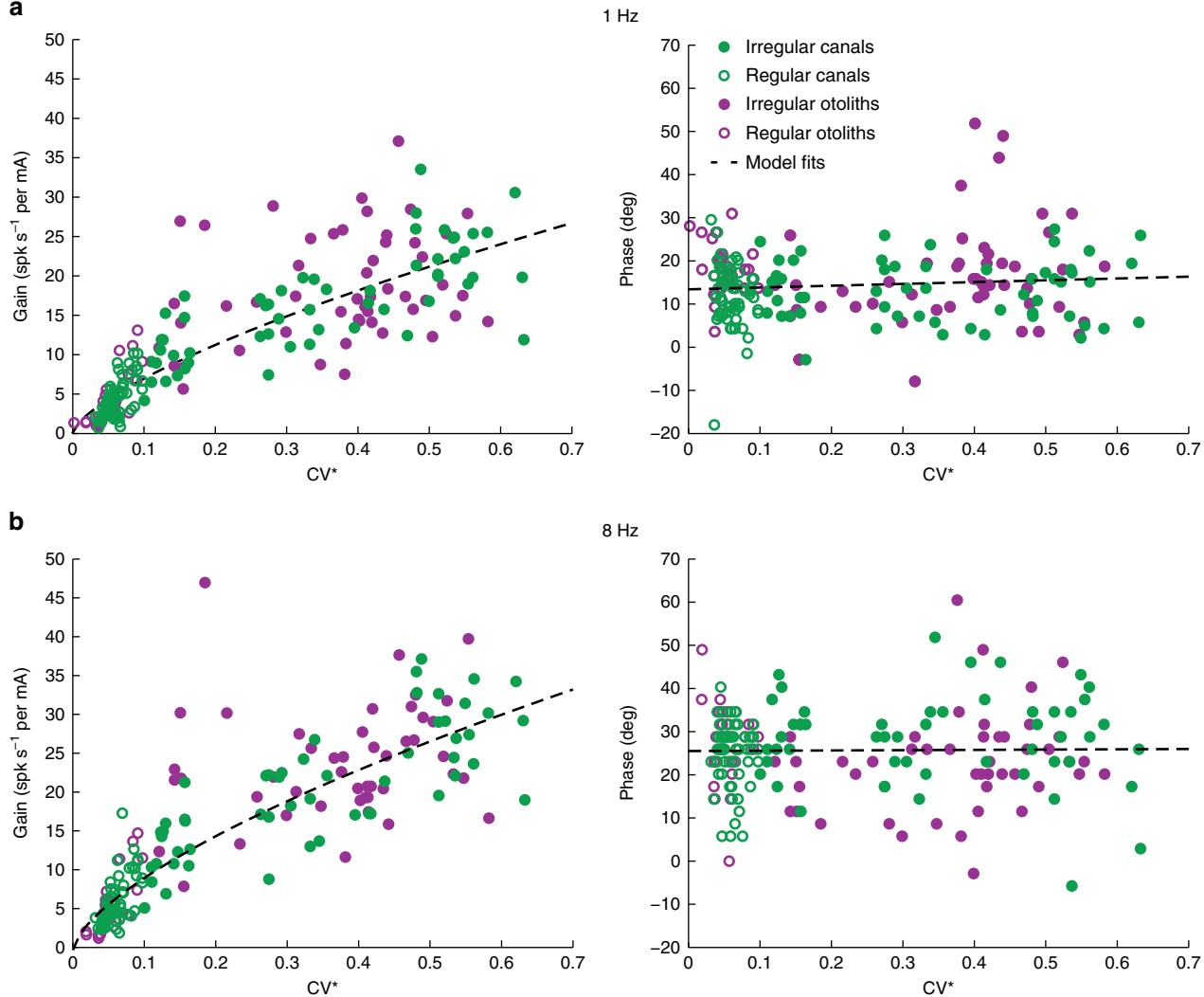

**Fig. 3** Comparison of canal and otoliths afferent responses to sinusoidal GVS. Gain (left) and phase (right) of canal (green circles, $N = 119$) and otolith (purple circles, $N = 84$) afferents to 1 Hz (**a**) and 8 Hz (**b**) sinusoidal stimulation as a function of the resting discharged variability, quantified as CV*. Dashed lines depict estimated power-law fits of gain (1 Hz: $a = 30.2$ [c.i. 27.1, 33.2], $b = 0.64$ [c.i. 0.6, 0.7]; 8 Hz: $a = 41.3$ [c.i. 37.7, 44.9], $b = 0.62$ [c.i. 0.6, 0.7]; left plots) and linear fits of phase (1 Hz: $a = 9.7$ [c.i. 4.8, 14.7], $b = 9.7$ [c.i. 8.3, 11.1]; 8 Hz: $a = 0.7$ [c.i. −7.2, 8.6], $b = 25.5$ [c.i. 23.2, 27.8]) to the combined canal and otolith afferent data set. c.i.: confidence interval

Bonferroni's correction for multiple comparisons) and regular (Student *t*-test: for 4–16 Hz, $p < 0.01$ Bonferroni's correction for multiple comparisons) canal afferents. Moreover, phase leads relative to stimulation were significantly higher for natural rotations versus GVS (Student *t*-test, irregular canal: all matching frequencies, $p < 0.0083$; regular canal: all matching frequencies, $p < 0.0083$ Bonferroni's correction for multiple comparisons).

Figure 5 illustrates an analogous comparison of otolith afferent responses to GVS versus natural motion stimuli. The example regular and irregular otolith afferents shown in Fig. 5a were typical in that the irregular afferent was more sensitive for translation (2 Hz) along its preferred direction (see Methods section). Quantification of our regular and irregular otolith afferent populations confirmed that average responses gains were comparable to those previously reported at 2 Hz for each otolith afferent class (Jamali et al.[45]; Student *t*-test: regular, $p = 0.81$ and irregular, $p = 0.16$). Next, to compare response dynamics of otolith afferents to GVS versus natural motion stimulation, we superimposed our data from Fig. 2e above (dashed lines) and data from the Jamali et al.[46] study (solid lines) which had quantified responses to translational motion

over the same frequency range. As was done above for the canal afferents in Fig. 4b, gains were normalized at 0.5 Hz to facilitate comparison across the two different stimulation conditions. Overall, the general trend of increasing gain and phase with frequency was similar for GVS- and motion-evoked responses in both otolith afferent types. Irregular otolith afferent responses, however, exhibited greater high-pass tuning for natural linear motion compared to GVS (Student *t*-test: for 8–16 Hz, $p < 0.01$ Bonferroni's correction for multiple comparisons). The normalized gains of regular otolith afferents, on the other hand, were comparable for natural motion and GVS (Student *t*-test: all frequencies, $p > 0.01$ ranges: 0.03–0.53; Bonferroni's correction for multiple comparisons). Finally, similar to irregular canal afferents, phase leads relative to stimulation were significantly higher for natural rotations versus GVS for irregular otolith afferents (Student *t*-test: for 2–8 Hz, $p < 0.0083$ Bonferroni's correction for multiple comparisons). Interestingly, however, this relationship was reversed for regular otolith afferent responses to GVS (Student *t*-test: for 0.5–8 Hz, $p < 0.0083$ Bonferroni's correction for multiple comparisons). Thus, together, the findings presented in Figs. 4 and 5 reveal the

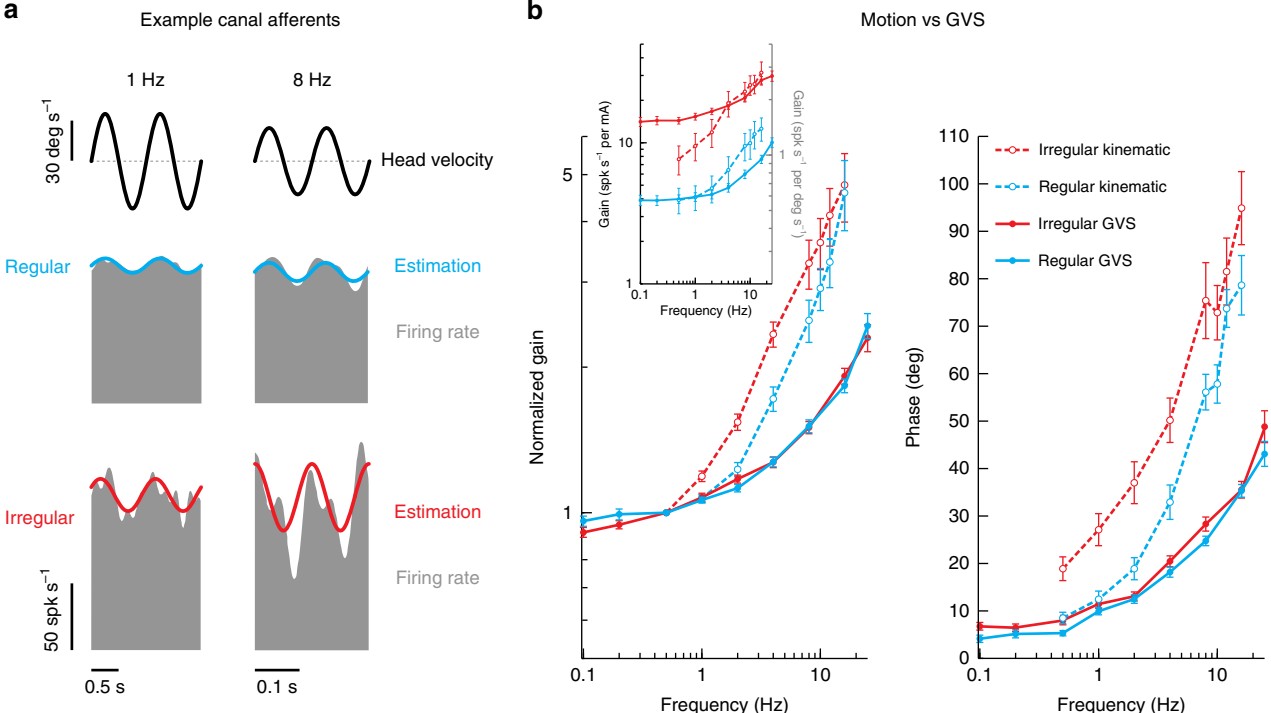

**Fig. 4** Comparison of motion and GVS for canal afferents. **a** Firing rate (gray) of example regular (blue) and irregular (red) canal afferents to 1 Hz (left) and 8 Hz (right) head rotational velocity. Firing rate estimates (solid blue and red line) were found using Eq. 4. The irregular canal afferent was more sensitive to rotation than the regular canal afferent (1 Hz: 1.05 versus 0.60 spk s$^{-1}$ per deg s$^{-1}$; 8 Hz: 2.20 versus 1.05 spk s$^{-1}$ per deg s$^{-1}$). **b** Population-averaged neuronal gain (i.e., sensitivity) (left) and phase (right) for regular (blue) and irregular (red) canal afferents as a function of the rotation (dashed lines) and GVS (solid lines) stimulus frequency. Gain in the main plot was normalized to responses at 0.5 Hz for comparison of responses across stimulus type, while the inset shows the non-normalized gain plotted with stimulus specific units. Error bars depict the s.e.m.

difference in high pass tuning features of canal and otolith afferent responses to GVS as compared to natural motion.

It is likely that the difference in the responses of canal and otolith afferents to these two types of vestibular stimulation occurs because GVS, unlike natural motion, bypasses the mechanotransduction of both end organs. Indeed, GVS directly activates the afferents (and potentially hair cells themselves) via electrical transmission. To test this hypothesis and more specifically evaluate whether the electrical properties of the afferents alone could account for their GVS-evoked responses, we used a simple model of vestibular afferent discharge developed by Smith and Goldberg[47] (see Methods section) to simulate regular and irregular afferent responses to the GVS applied in our experiments. Because the model relies on a voltage potential across the cell to simulate the applied current, we measured the voltage within the tissue surrounding the afferents to confirm that the simulated voltage could be represented as a scaled version of the applied current (see Methods section). In support of this assumption, we found a flat gain and phase across all frequencies between input current and measured voltage in neural tissue surrounding the vestibular nerve (see Fig. 6a—left box). This apparent ohmic property of the macaque head matches recent recordings in human cadavers, which similarly demonstrate negligible capacitive components[48]. We then scaled the simulated input voltage so that the model's spiking output matched the experimental data for both semicircular canal afferent classes. From the simulated afferent responses, we found that the model reproduced the frequency-dependent increase in gain and phase (see Fig. 6a—right box) as observed in our experimental results (see Fig. 6b). Thus, consistent with our hypothesis, a model based on the electrical properties of the afferents reproduced the main features of GVS-evoked afferent firing dynamics. We note,

however, that discrepancies between simulated and experimental responses were observed: (1) the rate of gain increase in the model was substantially lower, and (2) phases differed between regular and irregular afferents. This suggests that additional features not accounted for in the model (most notably hair cell contributions) may need to be included to fully replicate the experimental data.

To directly establish the correspondence between GVS-evoked and natural motion stimulation, we then estimated the average ratio of vestibular afferent sensitivity to GVS versus motion (either rotation or translation) at each frequency. Effectively, this motion-GVS ratio provides a measure of the equivalent virtual motion amplitude corresponding to a 1 mA current amplitude. Figure 7a, b show the population averaged motion-GVS ratios for canal and otolith afferents, respectively, as a function of frequency. The value of this ratio ranges from ~7 to 14 deg s$^{-1}$ per mA for canal afferents and 40–80 mG per mA for otolith afferents. Notably, irregular afferents generally displayed a frequency-dependent decrease in the motion-GVS ratio, which was well fit by an exponential function of the form $a \cdot e^{-b \cdot x} + c$ (red curves, Fig. 7a, b). Put another way, as frequency increased, higher GVS currents were required to achieve modulation corresponding to a given motion amplitude for both irregular canal and otolith afferents. In contrast, the motion-GVS ratio underwent relatively little change across the frequency range for regular canal afferents, while lower GVS currents were required as frequency increased to achieve afferent activity equivalent to a given motion amplitude for regular otolith afferents.

All together, our findings reveal for the first time the response dynamics of vestibular afferents firing activity evoked by GVS. Our data demonstrate the differences between the responses evoked by GVS versus natural motion stimuli, thereby providing

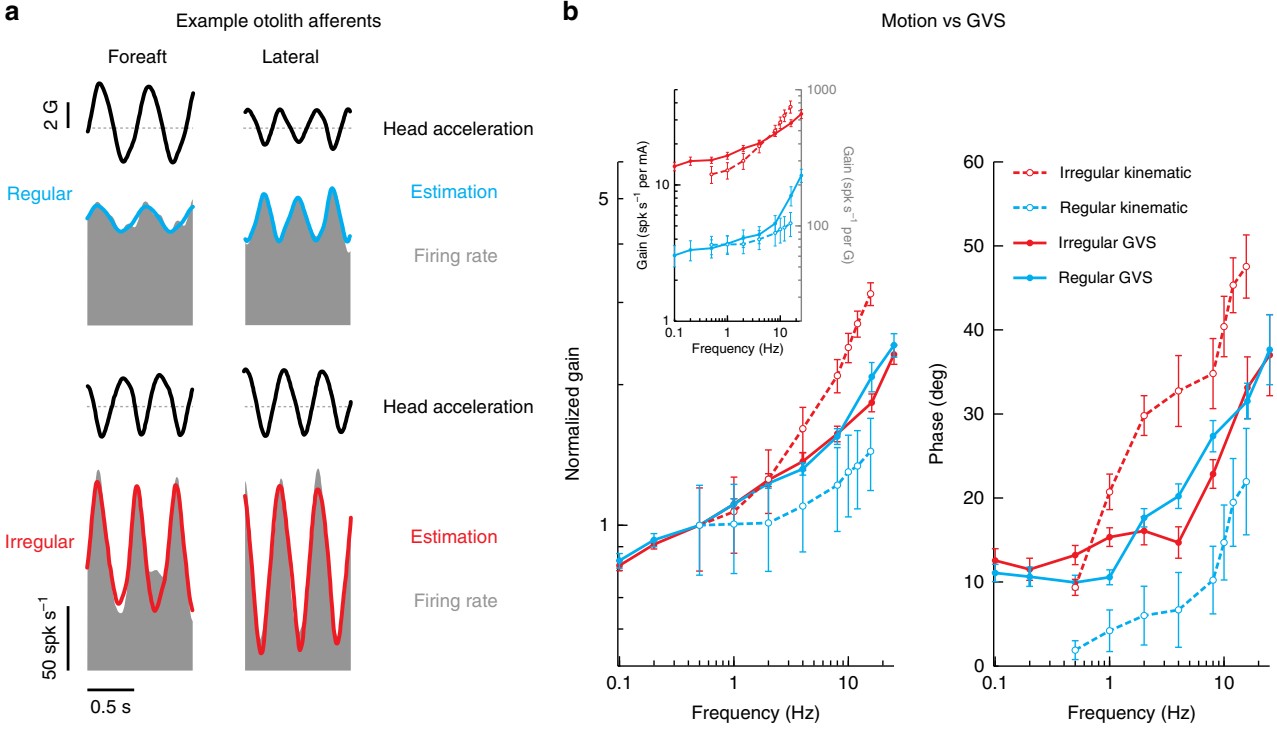

**Fig. 5** Comparison of motion and GVS for otolith afferents. **a** Firing rate (gray) of example regular (blue) and irregular (red) otolith afferents to ~2 Hz foreaft (left) and lateral (right) head acceleration. Firing rate estimates (solid blue and red line) are found using Eq. 5. The irregular otolith afferent was more sensitive to translation than the regular otolith afferent (2 Hz: 223.4 ± 31.2 spk s$^{-1}$ per G versus 68.6 ± 17.3 spk s$^{-1}$ per G). **b** Population-averaged neuronal gain (i.e., sensitivity) (left) and phase (right) for regular (blue) and irregular (red) otolith afferents as a function of the translation (dashed lines, from Jamali et al.[46]) and GVS (solid lines) stimulus frequency. Gain in the main plot was normalized to responses at 0.5 Hz for comparison of responses across stimulus type, while the inset shows the non-normalized gain plotted with stimulus specific units. Error bars depict the s.e.m.

a fundamental step to advance the applicability of GVS for biomedical uses in humans.

**Afferent detection thresholds do not vary with afferent type**. So far our analyses have focused on the computation of gain and phase measures to understand the dynamics of GVS-evoked afferent responses. However, the responses of vestibular afferents can also be quantified in terms of their signal-to-noise as a function of frequency[44,45,49]. To obtain a measure of signal-to-noise of GVS-evoked vestibular afferent responses, we next computed neuronal detection thresholds of single canal and otolith afferents. The detection threshold of an individual afferent was determined at each stimulation frequency by first plotting its time-dependent firing rate as a function of stimulus current. This is shown in Fig. 8a, b for example regular and irregular canal and otolith afferents during 1 Hz stimulation. Using signal-detection theory (see Methods section), we then compared the firing rate distribution for a given value of stimulus current to that obtained for zero current using the $d'$ measure (see Fig. 8c, d for the example canal and otolith afferents, respectively). The population averaged estimates of $d'$ spanned a limited range of ~0.4–0.6 mA across afferent type, discharge regularity and frequency (Fig. 8e, f). Detection thresholds were not significantly different when compared between regular and irregular afferents for either canals or otoliths (Student $t$-test: for all frequencies except canal afferents at 8 and 16 Hz, $p > 0.00625$ Bonferroni's correction for multiple comparisons). Further, thresholds were not significantly different when compared between canals and otoliths for either regular or irregular afferents (Student $t$-test: for all frequencies, $p > 0.00625$ Bonferroni's correction for multiple comparisons). We

did, however, observe variation in threshold values for all afferent types with increasing frequency (linear mixed model: $p < 0.001$): pairwise comparisons revealed that irregular afferent thresholds decreased by ~11–15% from 0.1 to 1 Hz and increased by ~25–33% from 1 to 16 Hz, while regular afferent threshold decreased by ~10–30% from 0.1 to 1 Hz but remained unchanged thereafter. Overall, our results show that primary vestibular afferent thresholds, and thus the signal-to-noise ratios, vary moderately with frequencies of GVS, but are invariant across the firing rate regularity and the end organ the afferent innervates. These results may have important implications in relation to human studies that have considered the perceptual consequences of GVS (see Discussion section).

## Discussion

In the present study, we recorded eye movements and the activity of primary vestibular afferents in nonhuman primates during electrical stimulation applied between surface electrodes placed on the mastoid processes, a setup analogous to human GVS studies. We first validated that sinusoidal GVS evoked torsional eye movements similar to those observed in human studies. We then characterized the responses of individual primary vestibular afferents to GVS at frequencies within the physiologically relevant range for natural head motion stimuli (0.1–25 Hz). Both semicircular canal and otolith afferents showed robust and comparable responses, characterized by tuning that showed monotonic increases in gain and phase lead as a function of frequency. In addition, we recorded the responses of the same canal and otolith afferents to physiological stimulation (i.e., motion) and found responses comparable to those reported in prior studies[45,49].

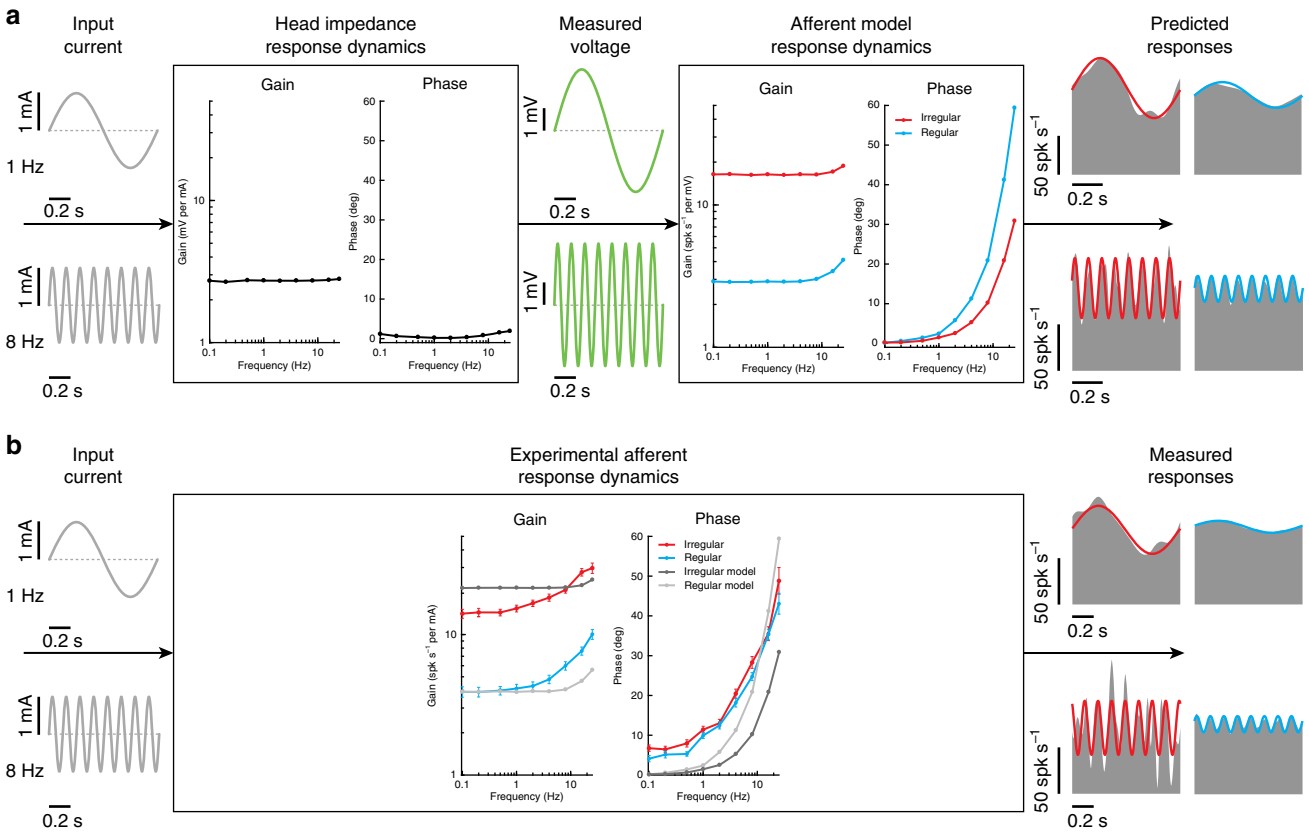

**Fig. 6** Simulation results of a vestibular afferent model to galvanic vestibular stimulation. **a** Current input (gray) produces voltage changes (green) in the tissue surrounding the afferent with a flat gain and phase at all frequencies (i.e., head impedance response dynamics—left box). These voltage changes were then used as input to the afferent model to produce the changes in afferent firing rate (gray) for example regular (blue) and irregular (red) afferents. The resultant afferent model response dynamics (right box) produce increasing gains and phases as a function of frequency for both afferent types.
**b** Experimentally estimated canal afferent response dynamics as a result of a current input are replotted here for comparison to the simulated data. Similar to the experimental responses, simulated data produced frequency dependent changes in gain and phase. For illustrative purposes simulated gain results were normalized to regular canal afferent responses at 0.1 Hz

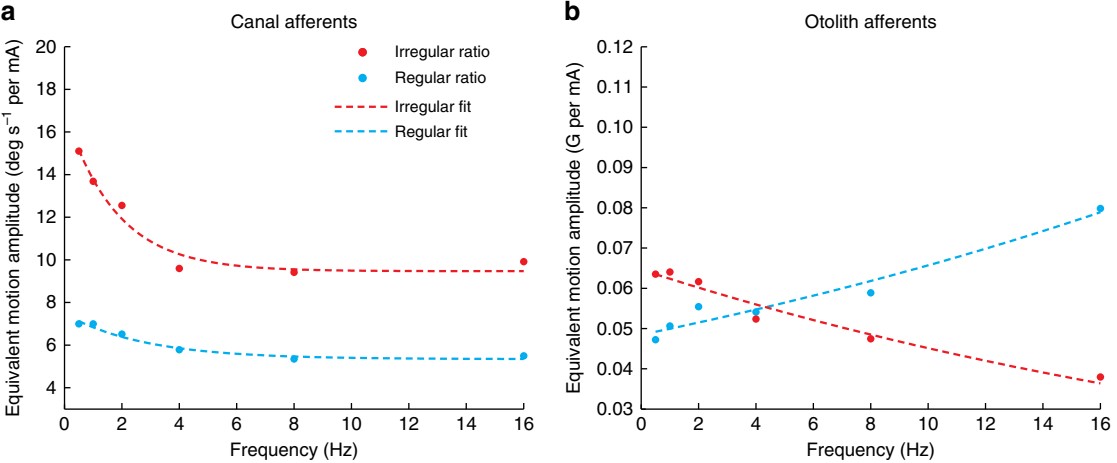

**Fig. 7** Motion-GVS ratios for vestibular afferent responses. **a** The motion-GVS ratio of both regular and irregular canal afferents decreases as a function of frequency. **b** The motion-GVS ratio of irregular otolith afferents decreases as a function of GVS frequency, but increases for regular otolith afferents. Specifically, for the same amplitude of current stimulation, the equivalent motion stimulation strongly decreased as a function of frequency for irregular canal afferents ($a = 7.55$ [c.i. 4.14, 10.96], $b = 0.56$ [c.i. 0.01, 1.11], $c = 9.49$ [c.i. 8.10, 10.89]) and otolith afferents ($a = 0.065$ [c.i. 0.061, 0.069], $b = 0.036$ [c.i. 0.025 0.047]). In contrast, this ratio was relatively constant for regular canal afferents ($a = 2.13$ [c.i. 1.35, 2.92], $b = 0.36$ [c.i. 0.01 0.71], $c = 5.36$ [c.i. 4.85, 5.88]), and actually increased as a function of GVS frequency for regular otolith afferents (blue curve, Fig. 6b; $a = 0.048$ [c.i. 0.044, 0.052], $b = -0.030$ [c.i. $-0.039$, $-0.022$]). For irregular and regular otolith afferents the c parameter was removed from the fit since confidence intervals (c.i.) overlapped zero. All model $R^2 > 0.94$–$0.98$)

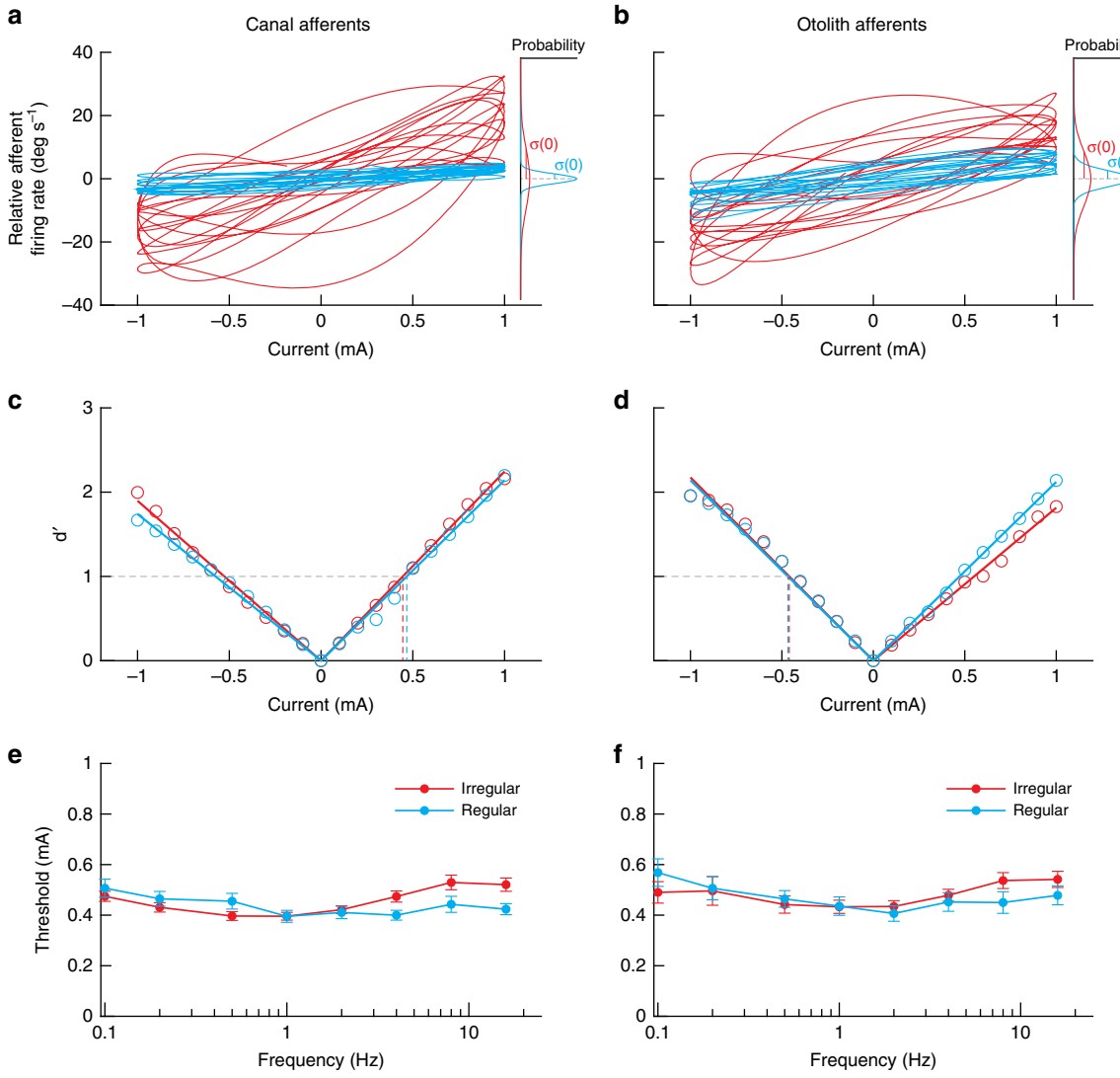

**Fig. 8** Detection thresholds for regular and irregular canal and otolith afferents during 1 Hz sinusoidal GVS. **a**, **b** Firing rate plotted as a function of stimulus current for regular (blue) and irregular (red) canal and otolith afferents. **c**, **d** Signal detection theory was used to compare the distribution of firing rates for a given value of stimulus current and that obtained for zero current (see insets in **a**, **b**) to obtain a plot of $d'$ as a function of stimulus current. The detection threshold $T_{d'}$ was computed as the lowest absolute value of stimulus current for which $d' = 1$ (dashed horizontal gray lines). **e**, **f** Population-averaged detection threshold values $T_{d'}$ for regular (blue) and irregular (red) canal and otolith afferents as a function of frequency. Neuronal detection thresholds varied with GVS frequency for all afferent types (linear mixed model: $p < 0.001$), but remained relatively unchanged across firing rate regularity and end organ innervation (Student $t$-tests: $p > 0.00625$ Bonferroni's correction for multiple comparisons). Error bars depict the s.e.m.

Specifically, afferents showed a difference in their response to GVS versus natural motion in that the high pass tuning was significantly greater for the latter. Using a simple stochastic model, we were then able to provide an explanation that largely accounted for the main features of afferent response tuning to GVS. Finally, we computed the neural detection thresholds of semicircular canal and otolith afferent responses to GVS. We found that the increased sensitivity of afferent responses to GVS matched the increased intrinsic variability of irregular versus regular afferents such that neuronal thresholds of primary vestibular afferents varied minimally with frequency, regardless of firing rate regularity or end organ innervation. Further, despite the lower sensitivity of regular afferents, they transmitted equivalent levels of information as irregular afferents to central vestibular pathways for detecting GVS-evoked sensations of self-motion. Together, our results reveal how GVS activates individual afferents at the first stage of vestibular processing, thereby establishing an essential step in the development of

physiologically accurate models of GVS required for further advancing the applicability of this non-invasive technique for biomedical uses in humans.

To determine how vestibular afferents respond to transmastoid GVS, we recorded from both canal and otolith afferents. Our findings reveal that irregular afferents in both end organs were substantially more responsive than regular afferents, consistent with previous animal studies, which delivered current inside the ear to stimulate vestibular afferents[33,35,41,42,50]. Moreover, we found that afferents from both classes of sensory end organs were equally responsive to GVS across the frequency range tested[35,36], and that the relationship between the afferent sensitivity and discharge variability followed a power law[33,36,42]. The response dynamics of vestibular afferents, however, differed from those previously reported for internal electrical stimulation[33,36,41]. Specifically, we found that transmastoid GVS evoked analogous changes in canal and otolith afferent responses as a function of frequency; response gains increased by over two-fold and phase

leads increased by ~30–40 degrees across the physiological frequency range (0.1–25 Hz). Furthermore, while irregular afferents were more sensitive to GVS than regular afferents, there was no difference in phase lead for irregular versus regular afferents. Thus, our results contrast with experiments using internal stimulation in squirrel monkey and chinchilla reporting far smaller increases in phase lead as a function of frequency[33,36]. In addition, these previous studies reported relatively constant response gains for regular afferents, while we observed a two-fold change in response gain with increasing frequency. A previous comparison of average change in afferent responses to constant current steps reported that external GVS is ~10 times less effective than internal stimulation in guinea pig head[35]. Given that the macaque monkey head is considerably larger, we expect an even greater difference in the two conditions and further speculate that such bioelectrical filtering could also contribute to differences between vestibular afferents responses to internal versus external GVS.

To understand the mechanisms underlying the observed tuning of vestibular afferents to transmastoid GVS, we considered two explanations. First, internally delivered current is applied in close proximity to vestibular afferents, while in the present study we applied GVS at a greater distance on the mastoid processes. External recording techniques, such as electroencephalography, have shown that the underlying skin, bone and cerebrospinal fluid together effectively function as a low-pass filter to the electrical activity generated in the brain[51]. Following the same principle, the application of GVS current to the skin surface of the mastoid should result in greater low-pass filtering of the stimulus before activating the vestibular hair cells and/or afferents than internal stimulation. Thus, currents delivered at higher frequencies should be attenuated leading to smaller gains and phase lags in the response of vestibular afferents to transmastoid as compared to internal current stimulation. This was, however, the opposite of what we observed; voltages measured in the surrounding tissue demonstrated ohmic properties of the head[48], and afferent responses generally displayed larger gain and phase leads with increasing frequency for transmastoid versus internal stimulation[33,36]. These results further emphasize that vestibular afferent dynamic responses to GVS cannot be predicted from previously reported responses to internal stimulation.

We next considered a second possible explanation, namely that the electrical properties of afferent spiking could largely explain their frequency tuning to transmastoid GVS. To test this, we simulated a model developed by Smith and Goldberg[47] to describe regular versus irregular vestibular afferents in monkey[42]. This simple model accounts for differences in the after-hyperpolarization (AHP) following each action potential and synaptic noise. Specifically, the AHP is made shallower and faster, and the synaptic noise is set higher for irregular versus regular afferents. Because of their higher synaptic noise, a depolarizing input is more likely to push the voltage of an irregular than regular afferent over threshold, resulting in an increased sensitivity to applied current. We evaluated this model—with parameters set to reproduce data from primates[47]. Consistent with our hypothesis, we found that the electrical properties of afferent spiking alone explains key features (increasing phase lead) of the afferent response tuning to transmastoid GVS in monkey (Fig. 6).

The remaining unexplained characteristics of the afferent responses (higher gains and phase leads) could suggest a role for other factors (i.e., hair-cell mediated activity) not currently incorporated in the model. Indeed, a recent study in Xenopus suggests that ~30% of the change in afferent firing is the result of direct hair cell activation[52]. This result is further consistent with the conclusions of a GVS study in patients with gentamicin-induced hair cell loss[53]. It is notable that hair cell activation in healthy subjects could also occur by retrograde GVS activation of nonquantal

transmission[54–56]. Since nonquantal transmission is faster than conventional quantal transmission, this mechanism could also potentially contribute to increased phase leads we observed relative to the Smith and Goldberg model. Additionally, more recent studies[57–61] indicate that the Smith and Goldberg model does not fully capture the dynamics of the conductances (e.g., $K_{LV}$, HCN, and SK) of vestibular afferents. The spike-making ion channels in afferents can contribute to generating higher gain and phase leads with increasing frequency[54]. For example, $K_{LV}$ conductances produce more negative spike thresholds resulting in increased phase lead[54]. These conductances are more densely expressed by irregular neurons[54,59,62], which in turn show greater phase leads. The development of a mechanistic model of GVS that accounts for both hair cell activation as well as the numerous conductances within the afferents will be an interesting and important direction for future work.

To advance the use of GVS for biomedical applications as well as virtual reality in humans, it is important to address the question of whether and how it is possible to equate natural motion and GVS-evoked afferent activity. Our results indicate that vestibular afferent response dynamics differ for GVS versus real motion stimulation. This finding is not unexpected given that GVS activates afferents by bypassing the biomechanics of both the semicircular canals and otolith organs, which contribute to the dynamics of responses to rotation and translation, respectively. The similar transfer functions of GVS-to-canal and GVS-to-otolith afferents, despite the different mechanics of the canals/ otoliths, suggest that the GVS-afferent transfer functions are dictated mainly by the properties of the afferents, their responses to the GVS-induced voltage (as predicted by the stochastic model of repetitive activity in vestibular afferent) and other factors including the various conductances within the afferents and vestibular hair cells recruitment by GVS. In support of this proposal, we further found that neuronal detection thresholds (i.e., signal-to-noise ratios) of vestibular afferents do not vary substantially with discharge regularity or the class of sensory end organ they innervate. Importantly, the similarity in detection thresholds across end organ innervation refutes the proposition from human behavioural responses that otolith afferents have lower thresholds to GVS as compared to canal afferents[3]. Furthermore, our results also suggest that, as a function of frequency, response sensitivity and variability increase in parallel for all afferent types. Interestingly, the invariance of thresholds to GVS across regular and irregular afferents matches neuronal thresholds of otolith afferents during natural motion[63], but contrasts those of canal afferents during motion, which are ~2–3 times higher in irregular versus regular firing afferents[44,49]. Overall, these results are of particular importance for human GVS studies, and suggest that despite the lower sensitivity of regular afferents to GVS compared to irregular afferents, they transmit equivalent information to central vestibular pathways for the detection of GVS-evoked sensations of self-motion.

Studies that have attempted to model GVS-evoked behavior in humans based on vestibular afferent responses have not considered vestibular response dynamics or their thresholds[1,7,64–67]. For example, two recent studies have assumed that GVS-evoked canal afferent responses have relatively flat tuning across frequency[66,67], as suggested by studies of internal stimulation applied in squirrel monkeys and chinchilla[33,36], while another assumed that the increased sensitivity of irregular over regular afferents makes them more suited to detect GVS-evoked sensations of self-motion[68]. Our present results, however, show that these assumptions are physiologically incorrect, and establish the natural relationship between motion-evoked and GVS-evoked vestibular responses (Figs. 2d, e and 8d, e). These relationships, in turn, provide the ability to compute the equivalent motion

corresponding to GVS-evoked afferent activity[69] that is required to develop models explaining response dynamics within vestibular pathways during GVS[70].

Our findings establish a valid nonhuman primate model for investigating the effects of GVS on vestibular afferents. Notably, sinusoidal GVS in our monkeys evoked torsional eye movements with similar response dynamics to those reported in humans[4,6,7]. Further, as in human studies[71], we observed variability in the gain of torsional eye movements among animals; for example, Monkey B exhibited greater torsional eye velocity compared to Monkey H. Importantly, this difference in the eye movement response sensitivity was linked to a difference in afferent response sensitivity between these animals. Thus, between-subject variability of GVS-evoked behaviors in humans may in part be explained by vestibular afferent responses to GVS differing between subjects. While it is possible that difference in thickness of the skull and/or overlying tissue contributed to the observe differences, we note that the average size of both monkeys was comparable (4.9 versus 5.5 Kg for monkeys H and B, respectively).

Despite the success of our experiments aimed at establishing how vestibular afferents respond to GVS—to understand the relationship between GVS-evoked afferent responses and behaviors like eye movements and postural responses, or higher-order functions like perception—it is important to emphasize one unresolved issue. Specifically, GVS induces simultaneous activation of primary afferents from all 5 vestibular sensory end organs on one side with concomitant inhibition of those on the contralateral side—an activation pattern with no physiological motion equivalent. Fitzpatrick and Day[1] proposed a model of the perceptual effects of GVS in humans in which all otolith and semicircular afferents are activated by the transmastoid electrical current, resulting in net cancellation of the otolithic signals at the population level and a net GVS-vector that predominantly reflects canal activation (see Appendix in Mian et al.[72]). Based on this model, Peters et al.[17] recently recorded the perceptual direction recognition thresholds in humans during real rotation and GVS from 0.1 to 2 Hz, and found that while thresholds for real rotation improve as a function of frequency, those for GVS worsen as a function of frequency (i.e., more current is need for higher frequencies). Our results show, however, a slight decrease (~10–30%) in neuronal thresholds across equivalent frequencies. The neuronal thresholds to GVS were also 30–60% lower than human direction recognition thresholds[17]. This contrasts with neuronal thresholds to natural motion, which are typically larger than human perceptual thresholds and require integration of multiple afferents by higher order brain areas to reach equivalent perceptual levels[45,49]. These differences between neuronal vs human perceptual thresholds to GVS and natural motion may be related to end organ morphology[73] as well as the larger head size and thicker bone of humans. Indeed, larger currents are required to evoke equivalent field potentials in brain tissue when applied to a human head compared to a rat head[48]. Furthermore, we speculate that the optimization of neural coding to the statistical structure of natural self-motion[74] influences the inherent electrical properties of vestibular afferent neuronal responses to GVS. This, however, is unlikely to contribute to the differences we observed between afferent thresholds in monkeys versus perceptual thresholds in humans since the statistics of natural motion in non-human primates and humans are comparable[75,76].

These considerations highlight that to link GVS-evoked vestibular afferent responses to the downstream behavior and perception, it will be essential to determine how central pathways integrate this unnatural vestibular afferent input. By establishing the neural integration processes of vestibular afferent activity during GVS, it may be possible to establish and confirm a similar estimate of the net GVS-evoked motion signal in macaque

monkeys. In turn, this knowledge could then explain GVS-evoked behavioral responses consistent with a net otolithic component[27,28] and/or the decline in GVS-evoked motion perception observed as a function of frequency[17], even though individually, vestibular afferents show limited frequency-dependent changes in threshold.

## Methods

**Surgical procedures.** Three male macaque monkeys (2 *Macaca fascicularis, Monkey B and H, and 1 M. mulatta, Monkey D*) were prepared for chronic extracellular recording using aseptic surgical techniques. All three monkeys were between 6 and 10 years of age and weighed 4.9–6.5 kg. The experimental protocols were approved by the McGill University Animal Care Committee and were in compliance with the guidelines of the Canadian Council on Animal Care.

The surgical preparation for Monkey B and D followed the procedures described previously[77]. Under a new protocol, Monkey H was administered loading doses of carprofen (4 mG kg$^{-1}$ sq) and cefazolin (22 mG kg$^{-1}$ iv), the latter of which was administered slowly and repeated every two hours for the duration of the surgery, to reduce swelling and prevent infection, respectively. In all three animals, aseptic surgical techniques were used. Under isofluorane anesthesia (0.8–1.5%), we secured a stainless steel post to the animal's skull with stainless steel screws and dental acrylic, permitting complete immobilization of the animal's head during the experiment, and implanted a chamber for chronic extracellular recording of vestibular nerve afferents. In addition, an eye coil, consisting of three loops of Teflon-coated stainless steel wire, was implanted in the right eye behind the conjunctiva. The post-surgery protocol followed for Monkeys B and D has been previously described[77]. For Monkey H, carprofen (2 mG kg$^{-1}$) administration was continued daily for 5 days, and buprenorphine (0.01–0.02 mg kg$^{-1}$ im) was administered postoperative for analgesia every 12 h for 2–5 days, depending on the animal's pain level. In addition, cefazolin (22 mG kg$^{-1}$ im) was injected twice within 24 h after surgery. All animals were given at least 2 weeks to recuperate from the surgery before any experiments began.

**Data acquisition.** During the experiments, monkeys were head-restrained and seated comfortably in a primate chair mounted on top of a motion platform (Space Control, France). The left vestibular nerve was found as previously described[45]. Extracellular single-unit activity of primary vestibular afferents (semicircular canal and otolith) was recorded using tungsten microelectrodes (7–10 MΩ; Frederick-Haer Co., Bowdoinham, ME). Neural signals were band-pass filtered from 300 Hz to 3 kHz and sampled at 30 kHz. Head linear acceleration and angular velocity was measured by a three-dimensional linear accelerometer and a one-dimensional angular gyroscope (Watson Inc., Eau Claire, WI), respectively, both firmly secured to the animal's head post. Horizontal and vertical eye positions were measured using the magnetic search coil technique[77].

Angular head velocity, linear head acceleration, eye position, and galvanic vestibular stimulation signals were low-pass filtered at 250 Hz (eight-pole Bessel filter) and sampled at 1 kHz. Neural, behavioral and stimulation data were collected through the Cerebus Neural Signal Processor (Blackrock Microsystems, Salt Lake City, UT). Neural data were imported into either Offline Sorter (Plexon, Dallas, TX)[78] or into a custom-written algorithm in MATLAB (The MathWorks, Natick, MA) to extract action potentials.

**Experimental paradigms.** Three dimensional eye position was measured using a modified eye tracker (200 Hz; Chronos Vision, Berlin, Germany) fixed onto the monkey's head post. GVS-evoked eye movements were recorded in the dark, while the monkey fixated a target spot projected on a cylindrical screen. Offline analysis software Iris (Chronos Vision, Berlin, Germany) was used to calculate torsional eye position from markers applied near the limbus. Markers consisted of an infrared absorbing cosmetic pigment, Eisenoxid 316/Schwarz (Carl Jäger Tonindustriebedarf GmbH, Erlen, Germany), dissolved in distilled water and were applied near the limbus using a sterile surgical marking pen. Search coil data were used to confirm that horizontal and vertical eye position recordings were similar across measurement platforms.

Once an afferent was isolated, we determined which vestibular end organ was its source of activation by recording neuronal responses to yaw rotations and translations in the horizontal plane. To assess that semicircular canal afferents recorded in this study were similar in sensitivity to previous studies, a subset of horizontal afferents were stimulated with yaw sinusoidal rotation at frequencies 0.5, 1, 2, 4, 8, 10, 12, and 16 Hz with peak velocity of ~40 deg s$^{-1}$. Similarly, otolith afferents were stimulated with translation in the fore-aft (90°) and lateral (0°) axes at ~2 Hz. Because our motion platform was unable to move the monkey in the vertical direction, afferents that were predominantly sensitive to linear stimulation along the vertical axis were not included in our dataset.

Galvanic vestibular stimulation was applied to animals using carbon rubber electrodes (~6 cm²) in a binaural bipolar configuration. The electrodes were coated with Spectra 360 electrode gel (Parker Laboratories, Fairfield, NJ) and secured over the animal's mastoid processes using a small bandage in a manner similar to human studies[70,79]. Stimuli were generated using MATLAB and were delivered as analog signals to a constant current isolation unit (STMISOLA; Biopac Systems

Inc., Goleta, CA) via a QNX-based real-time data acquisition system or an arbitrary waveform generator (Keysight Technologies, Santa Rosa, CA). The convention for the current polarity of the stimulation was set relative to the polarity of the left stimulating electrode, which was on the same side of the vestibular afferents recorded. In the figures, cathodal and anodal currents are depicted as positive and negative values, respectively. This convention was chosen to match and allow comparison with most human volunteer studies (anode-right, cathode-left as a positive signal for binaural bipolar GVS). To confirm that the voltage could be represented as a scaled version of the applied current, we recorded the voltage potential within the tissue surrounding the afferents during GVS at each frequency. For neural recordings, animals were exposed to a series of sinusoidal current (sinusoidal GVS) of frequencies 0.1, 0.2, 0.5, 1, 2, 4, 8, 16, and 25 Hz with peak amplitude of 1 mA. To test eye movement responses to sinusoidal GVS, animals fixated a laser spot target in darkness, while we applied sinusoidal stimulation at the same frequencies from 0.5 to 8 Hz, with a peak amplitude of 1 mA. Further, to test whether responses were linear, we applied 2 Hz sinusoidal currents of different amplitudes (i.e., 0.5, 0.75, 1, 1.25, and 1.5 mA, peak-to-peak amplitude). By convention, rightward horizontal, upward vertical and clockwise torsional (i.e., toward the right ear) eye movements are expressed as positive values.

**Data analysis.** All data were imported into MATLAB for analysis using custom-written algorithms. Angular head velocity, linear head acceleration, and eye position signals were digitally filtered with zero phase at 125 Hz using a 51st-order finite-impulse-response (FIR) filter with a Hamming window.

Afferents were first classified based on the regularity of resting discharge, which was assessed by computing the normalized coefficient of variation (CV*) as previously described[42,44]. Afferents with CV* ≤ 0.1 were considered as regular, while those with CV* > 0.1 were considered as irregular[80]. The afferents' average resting discharge, in the absence of movement or stimulation, was also computed. To estimate the time-dependent firing rate FR(t), we first assigned the spike train R(t) as the binary sequence of action potentials with bin width of 1 ms. Then, R(t) was convolved with a Kaiser window whereby the cut-off frequency was set to 0.1 Hz above twice the sinusoidal stimulus frequency to obtain the estimated FR(t)[81].

For each afferent, response dynamics to sinusoidal GVS were estimated at each stimulation frequency using a least-squares regression of the equation:

$$FR(t) = gain \times GVS(t + \theta) + bias, \quad (1)$$

where FR(t) is firing rate, gain is the afferent sensitivity to the sinusoidal GVS, $\theta$ is the phase shift relative to the GVS waveform and bias is an offset representing resting discharge. A minimum of ten cycles were included in the fit at each frequency. Next, to characterize the response dynamics of the afferent modulation resulting from GVS, linear time invariant models consisting of two poles and two zeros were estimated for both regular and irregular afferents originating from the canals:

$$H_1(f) = k\frac{(s + b_1)(s + b_2)}{(s + a_1)(s + a_2)}, \quad (2)$$

where $s = i2\pi f$. Parameter values were estimated from the population averaged frequency responses to sinusoid GVS using a least-squares regression to minimize the difference between the estimated and measured transfer functions in the complex form. For otolith afferents, we further found that more representative fits could be obtained using the fractional order transfer function described previously[82]:

$$H_2(f) = k\frac{s^{k_1}(1 + bs)^{k_2}}{(1 + as)}. \quad (3)$$

For each canal afferent, response dynamics to angular head velocity were estimated at each stimulation frequency using a least-squares regression of the equation:

$$FR(t) = gain \times \dot{H}(t + \theta) + bias, \quad (4)$$

where FR(t) is firing rate, gain is the afferent sensitivity to the head angular velocity $\dot{H}$, $\theta$ is the phase shift relative to the head velocity waveform and bias is an offset representing resting discharge. A minimum of ten cycles were included in the fit at each frequency. Because the head rotations for the vertical canal afferents were not in the plane of the canal, trigonometric corrections were made based on the canal planes in this species as previously described[43]. For each semicircular canal afferent, the decomposed angular velocity of the appropriate plane was then used to estimate the corrected gain.

For each otolith afferent, response dynamics to translational head acceleration were estimated at each stimulation frequency using a least-squares regression to the equation:

$$FR(t) = gain \times \ddot{H}(t + \theta) + bias, \quad (5)$$

where FR(t) is firing rate, gain is the afferent sensitivity to the head translational acceleration $\ddot{H}$, $\theta$ is the phase shift relative to the head translational waveform and bias is an offset representing resting discharge. A minimum of ten cycles were included in the fit at each frequency. Then for each frequency, the maximum gain and preferred direction was estimated using a cosine fit[83].

Similar to the analysis of afferent responses, least-squares regression analysis was used to determine the gain and phase shift of the eye velocity relative to sinusoidal GVS waveform. This analysis was performed on ≥5 cycles of desaccaded eye movement for each frequency of stimulation. Torsional eye velocity gains measured at each frequency and amplitude of stimulation were normalized relative to values computed at 0.5 Hz and 0.5 mA, respectively. Reported values were averaged across five trials.

**Numerical afferent model.** We used a vestibular afferent model developed by Smith and Goldberg[47] to assess whether the electrical properties of the afferents could account for the GVS-evoked responses. The afferent model captures the discharge regularity (i.e., CV*) of regular and irregular afferents by replicating their specific afterhyperpolarization properties and synaptic noise using the equation:

$$V(t) = \frac{g_S V_S + g_K(t) V_K + V_P}{1 + g_S + g_K(t)}, \quad (6)$$

where V(t) is the membrane potential, $g_S$ is the membrane conductance, $V_S$ is the excitatory synaptic potential ($V_S = 70$ mV), $g_K(t)$ is the afferent potassium conductance, $V_K$ is the potassium potential ($V_K = -30$ mV) and $V_P$ is the galvanic input. The membrane conductance is assumed to be a homogeneous shot-noise process composed of quantal events of rate $\lambda$, amplitude $\Delta g_S$ and effective duration $\Delta t_S$, such that the quantal EPSP size, measured at rest, is $A = \Delta g_S V_S$ for each afferent type (i.e., regular or irregular). The time dependent potassium conductance following an isolated spike is described by $g_K(t) = g_{K0} e^{(-t/\tau_K)}$, where t is the postspike time, $g_{K0} = g_K(t = 0)$ and $\tau_K$ is a time constant for each afferent type. The model assumes a cumulative summation of afterhyperpolarizations using a definite proportion ($p = 1$) of the $g_K$ left over from the preceding activity that was added to the $g_K$ triggered by each spike. Firing occurs when V(t) is greater than or equal to a fixed threshold $V_T$ (10 mV). The vestibular afferent model was simulated in MATLAB for the duration of the experimentally applied vestibular signals with a discrete time step of 0.1 ms.

We simulated a regular and irregular afferent with CV*s equivalent to the means obtained from our canal afferent data (regular = 0.06; irregular = 0.35) by setting the model parameters $g_{K0}$, $\tau_K$, and A similar to the approach described by Smith and Goldberg[47]. For the regular neuron, parameter values were: $g_{K0} = 1.90$, $\tau_K = 5.99$ ms, and $A = 0.168$ mV. For the irregular neuron, parameter values were: $g_{K0} = 0.60$, $\tau_K = 3.12$ ms, and $A = 0.779$ mV. The spiking responses from the model were analyzed in the same way as the experimental data.

**Neuronal threshold calculation.** To compute the neuronal threshold of an individual afferents for a given stimulus frequency, we first plotted its time-dependent firing rate as a function of the shifted stimulus GVS(t − θ) to obtain the instantaneous firing rate-stimulus current curve (see Fig. 8a, b). We did this using the Kaiser filter based approach[81]. Next, using a stimulus current bin width of 0.1 mA, we computed the mean and variance of the corresponding firing rate distribution for each stimulus current value (see Fig. 8a, b, insets). The neuronal thresholds were computed using the d′ measure from signal-detection theory[84], which assumes that the firing rate distribution is normal:

$$d'(GVS) = \frac{|\mu(GVS) - \mu(0)|}{\sqrt{(\sigma^2(GVS) + \sigma^2(0))/2}}, \quad (7)$$

where $\mu(GVS)$ and $\sigma^2(GVS)$ are the mean and variance of the firing rate distribution at stimulus current GVS, and $\mu(0)$ and $\sigma^2(0)$ are the mean and variance of the firing rate distribution at zero stimulus current, respectively. The d′ values were then plotted as a function of stimulus current and fitted with a first-order polynomial (see Fig. 8b). The detection threshold $T_{d'}$ was computed as the minimum of the absolute value of the positive and negative values of stimulus current for which d′ = 1[49]. Our analysis did not include 25 Hz stimuli because we were unable to extract data at the 0 mA bin due to the recording frequency.

**Statistical analysis.** Our sample sizes were similar to those generally employed in the field[45,49]. Statistical analysis was performed in SPSS (IBM, Armonk, NY) and Excel (Microsoft, Redmond, WA). Statistical significance was set at p < 0.05. Before statistical analysis, normality of distributions was evaluated using a Shapiro–Wilk's test. To analyze the eye movements in response to sinusoidal GVS, a repeated measures ANOVA was conducted with stimulation frequency or current amplitude as the within factor. To analyze the gain and phase of vestibular afferents as a function of frequency for canal and otolith afferents, two-tailed Student's t-tests were used to compare regular versus irregular responses at each frequency with a Bonferroni's correction for multiple comparisons. To analyze the relationship of the gain and phase of vestibular afferent responses to sinusoidal GVS as a function of CV*, power-law and linear regressions were performed on gain and phase data, respectively, for canal and otolith afferents separately at each stimulation frequency. If the 95% confidence intervals (c.i.) for the estimated parameters from the canal and otoliths were overlapping, a single estimate using all afferents was performed. When comparing afferent responses to GVS and motion, we estimated the average ratio of vestibular afferent sensitivity to GVS versus motion (either rotation or translation) at each frequency. Finally, to analyze the detection thresholds, we first natural-log transformed the data to satisfy normality. Student t-

tests were used on the transformed data to assess the effect of afferent regularity and end organ at each frequency with a Bonferroni's correction for multiple comparisons. In addition, variation in thresholds across frequency was assessed for each afferent class using linear mixed models on the transformed data. All values are expressed as mean ± s.e.m. unless otherwise specified. The detailed results of the statistical comparisons are listed in the Supplementary Table 1.

## Data availability
All data supporting the findings of this study have been deposited on Figshare under the https://doi.org/10.6084/m9.figshare.7718675.

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

## Acknowledgements

P.A.F. received funding from the European Union's Seventh Framework Programme for Research, Technological Development and Demonstration under the People Programme (Marie Curie Actions Grant 624158) and the Netherlands Organization for Scientific Research (NWO #016.Veni.188.049). J.-S.B. was funded by the Natural Sciences and Engineering Research Council of Canada (RGPIN: 356026-13). K.E.C. received funding from the National Institute on Deafness and Other Communication Disorders at the National Institutes of Health (grants R01-DC002390 and R01-DC013069) and the Canadian Institutes of Health Research. We thank Dale Roberts for his help with development of an approach to make direct voltage potential recordings during the application of GVS.

## Author contributions

Conceptualization, A.K., P.A.F., J.-S.B and K.E.C; Formal analysis, A.K. and P.A.F; Investigation, A.K. and D.E.M.; Writing—Original Draft, A.K., P.A.F. and K.E.C.; Writing—Review and Editing, A.K., P.A.F., D.E.M., J.-S.B., and K.E.C.

## Additional information

**Competing interests:** The authors declare no competing interests.

