## [Peer Review File · Nature Communications]

Reviewers' Comments:

Reviewer #1:

Remarks to the Author:

In their study "Neural Substrates and Dynamics of Galvanic Vestibular Stimulation in the Behaving Primate" Kwan and colleagues investigate single vestibular afferent and eye movement responses to transmastoid galvanic vestibular stimulation (GVS) compared to actual head motion stimulation in three awake-behaving macaque monkeys. Their major findings are that (1) GVS activates both primary vestibular afferents in the semicircular canals and otolith organs, (2) irregular afferents exhibit a higher sensitivity to GVS compared to regular afferents, and response dynamics of vestibular afferents to GVS across different frequencies markedly differ from responses to natural motion stimulation. Experimental procedures are sound and the data analysis appears robust. The findings are important and of broad interest since they reveal in a non-human primate model the neural activation pattern of GVS, a method that has gained increasing interest in probing the human vestibular system in health and disease.

Majors comments:

(1) My primary concern is related to the novelty of the present findings. Other than the authors suggest, vestibular afferent response patterns to transmastoid GVS have been studied previously in mammals (see Kim and Curthoys, *Brain Res Bull*, 2004). The findings of this earlier study are largely consistent with the present results, namely that GVS in parallel activates primary afferents in the semicircular canals and otolith organs and that the sensitivity of vestibular afferents to GVS depends on their firing rate regularity in a power-law relationship with irregular afferents being more sensitive to GVS than regular ones.

(2) As the authors mention, the tuning across different frequencies of afferent responses to transmastoid GVS differs markedly from that to internal GVS. What could be the reason for this difference? A direct comparison of internal vs. external GVS (as performed in Kim and Curthoys, *Brain Res Bull*, 2004) would be required to elucidate this issue in detail. The model of vestibular afferent discharge employed by the authors can apparently not account for this difference. The authors speculate that one reason for this difference could be that vestibular hair cells and afferent fibers are differentially recruited by internal vs. external GVS. What is the rationale for this assumption?

(3) A recent study indicates that GVS does not bypass vestibular hair cells but indeed activates them in a frequency-dependent manner (see Gensberger et al., *J Neurosci*, 2016). How could this piece of information be incorporated into the model simulations employed in the present study to more adequately estimate the response pattern of vestibular end-organs to GVS?

(4) Why did the authors not compare eye movement responses to GVS vs. natural motion stimulation as in previous studies (e.g. Gensberger et al., *J Neurosci*, 2016). This would provide important information whether the frequency-dependent responses of vestibular afferents to GVS vs. natural motion stimulation are preserved at later stages of vestibular information processing.

Minor comments:

(1) It is not clear from the methods section what eye tracking technique was used to record eye movement responses to GVS. Was it search coil, video-oculography or both?

(2) What were the exact limitations of the experimental setup that precluded linear motion stimulation along the vertical axis?

(3) Please correct: "To test eye movement recordings to sinusoidal GVS ..." (line 503)

(4) Please correct: "The motion-GVS ration of both regular and irregular canal afferents, as well as of irregular afferents, decreases as a function of frequency ..." (lines 863-865)

Reviewer #2:

Remarks to the Author:

Galvanic vestibular stimulation (GVS) has been used in many studies in humans as a means of activating the vestibular system without having to physically move the head and/or body. A major limitation of these studies is that it has been almost completely unknown how GVS induces activity in the vestibular system, thus making its effects on behavior difficult to interpret. This study from the Cullen lab applies GVS to monkeys and records the activity of vestibular afferents to both GVS and physical motion stimuli. While this is only a first step in understanding how GVS activates the vestibular system, it is a very important first step. Notably, the results show that GVS similarly activates both canal and otolith afferents, which is not what had been assumed in many studies. Thus, this study presents a very important advance for understanding how GVS impacts vestibularly-driven behavior.

Overall, I find this to be a well-conducted study that is a valuable contribution to the literature. I have a couple of substantive concerns that should be addressed, as well as a few minor issues.

Major comments:

1) I was a bit disappointed that the authors only characterized neural responses in terms of gain and phase. It would also be quite valuable to know the effect of GVS on neural sensitivity across frequencies. As the Cullen lab has shown previously very nicely, the selectivity profile of response gain and sensitivity (signal/noise ratio) are not necessarily the same. In order to help these findings be interpreted in terms of effects of GVS on behavioral sensitivity, it would also be quite useful for the authors to quantify neuronal sensitivity and present these results also.

2) I found there to be some tension and inconsistency among the way that the authors frame and state some of their conclusions. In comparing responses to GVS and real motion stimuli (e.g., Figs. 4&5), the authors repeatedly describe the results as "marked differences" between GVS and motion. On the other hand, when comparing measured responses to GVS with the model predictions (Fig. 6), the authors describe these differences as "minor discrepancies". To me, the differences between data and model in Fig. 6 look just as large as the differences between responses to GVS and motion, so these characterizations seemed rather arbitrary, non-rigorous, and somewhat self-serving. If it is important to make such distinctions, then the authors will need to quantify the deviations between datasets appropriately and make these comparisons quantitatively. I would argue that the differences in frequency tuning between GVS and real motion stimuli are not as dramatic as the authors seem to want to make them seem; I was actually surprised that they are as similar as they are.

Minor comments:

3) P. 7: I found the reporting of the (non-significant) statistics here, e.g., " $p > 0.0056$, Bonferroni's correction...", to be unsatisfactory. This doesn't give the reader any clear sense of how close or far these results are from being significant. This is the reason that there is a push toward reporting exact p values recently.

4) Lines 341-343: the last sentence of this paragraph is not a proper sentence grammatically

5) Line 550: there is no 'gain' variable in Equation 5

Reviewer #3:

Remarks to the Author:

This paper offers a very useful addition to previous work of Goldberg and colleagues. Rather than applying current invasively within the tissue, here it is applied cutaneously to characterising the transfer function between an externally applied current and the resulting change in vestibular nerve activity. The ultimate rationale is to better understand the effects of GVS upon human vestibular nerve activity, where the technique is limited to cutaneous stimulation. The methods and approach seem sophisticated and sound. I have a few queries below. In particular, the authors should highlight any limitations/differences in applying these findings to humans, since this is a key point of the paper. For example, the transfer function of GVS may be tuned to different peripheral mechanics in monkeys eg smaller canals etc.

- Line 108-9: You mentioning that you mainly saw torsional eye movement, and that this agrees with prior studies. However, this is slightly disingenuous because, as you state in the figure legend/methods, the monkeys were fixating on a light which would suppress any vertical/lateral movement. For transparency, I suggest you mention the fixation light here in the results text too.
- Figure 1C (inset) & Discussion: The authors point out that the absolute gain is quite a bit higher for Monkey B. They speculate anatomical differences, but don't say whether they looked for this in the monkey e.g. Did this monkey have a smaller head/thinner skull?
- Figure 4 shows that neural response gain increases with frequency for both GVS and real motion. This is perhaps surprising – would you not expect the two stimuli to show opposing effects? i.e. the GVS-nerve transfer function might be expected to compensate for the mechanics of the canals/otoliths? Maybe my thinking is overly simplistic, but nevertheless this could do with some discussion.
- Legend applying to figure 4b: Here it states values of 0.3 vs 0.1 spk/s/deg/s at 1Hz. However, the inset graph values in 4b do not seem to tally with these values, where the minimum value seems to be roughly 0.5 spk/s/deg/s. Please clarify.
- Legend figure 7: line 865 – Do you mean 'decreases' rather than 'increases'?
- Figure 6A: I might have expected tissue capacitance to produce changes in gain and phase with frequency, so the flat gain/phase plots of the Head Impedance Dynamics are a little surprising. Can you speculate whether the much larger head of a human being might introduce such capacitive filtering effects, not seen in the monkey?

Reviewers' comments:

Reviewer #1 (Remarks to the Author):

In their study “Neural Substrates and Dynamics of Galvanic Vestibular Stimulation in the Behaving Primate” Kwan and colleagues investigate single vestibular afferent and eye movement responses to transmastoid galvanic vestibular stimulation (GVS) compared to actual head motion stimulation in three awake-behaving macaque monkeys. Their major findings are that (1) GVS activates both primary vestibular afferents in the semicircular canals and otolith organs, (2) irregular afferents exhibit a higher sensitivity to GVS compared to regular afferents, and response dynamics of vestibular afferents to GVS across different frequencies markedly differ from responses to natural motion stimulation. Experimental procedures are sound and the data analysis appears robust. The findings are important and of broad interest since they reveal in a non-human primate model the neural activation pattern of GVS, a method that has gained increasing interest in probing the human vestibular system in health and disease.

Major comments:

(1) My primary concern is related to the novelty of the present findings. Other than the authors suggest, vestibular afferent response patterns to transmastoid GVS have been studied previously in mammals (see Kim and Curthoys, Brain Res Bull, 2004). The findings of this earlier study are largely consistent with the present results, namely that GVS in parallel activates primary afferents in the semicircular canals and otolith organs and that the sensitivity of vestibular afferents to GVS depends on their firing rate regularity in a power-law relationship with irregular afferents being more sensitive to GVS than regular ones.

We thank the reviewer for this question, and have revised the text to make the novelty of our findings relative to the Kim and Curthoys 2004 study more clear.

First, we agree our current findings in macaque monkey showing that (i) afferents from all end organs are activated by external GVS stimulation, and (ii) afferents sensitivity to GVS increase with discharge variability (CV*), are consistent with the previous work of Curthoys and colleagues. Indeed, this point was emphasized in our original manuscript. Importantly, however, as detailed in our Introduction, induced behavioural responses (e.g., eye movements) in prior GVS in human have led to conflicting interpretations whether GVS-evoked vestibular responses are predominately driven by the activation of the semicircular canals or otoliths. Thus, the development of an alert macaque based model, in which we can record eye movements, first provides a novel opportunity to directly address this open question.

Furthermore, a second key advance of our study is that, unlike the Kim and Curthoys study, our experiments were designed to directly assess the response dynamics of vestibular afferents firing activity evoked by GVS. In their 2004 study in anesthetized guinea pig, Kim and Curthoys' solely measured the mean change in afferent firing rate produced by a current step. Accordingly, this previous study only described how averaged afferent activity changes during the step, and did not characterize response dynamics or consider differences between motion-evoked versus GVS-evoked response dynamics.

Finally, a third novel and important contribution of our study is that we report, for the first time, the neural detection thresholds of vestibular afferents responses to GVS. Specifically, in response to Reviewer 2's comments, we now compute and compare the neural detection thresholds of regular

versus irregular afferents, for the canal and otolith systems. This measure is effectively a measure of signal-to-noise, which provides a significant advance for the interpretation of recent studies focused on understanding the perceptual consequences induced by GVS. Accordingly, we now consider the implications of our findings relative to this literature in the revised Discussion. Overall, we find that the detection thresholds of primary vestibular afferent thresholds vary minimally with the frequency of GVS, the firing rate regularity or the end organ the afferents innervate. This indicates that changes in sensitivity to GVS overcome any adverse effects of increased intrinsic neuronal variability, such that all afferents transmit equivalent levels of information to central vestibular pathways to detect GVS-evoked sensations of self-motion.

We have revised the Introduction, Results and the Discussion covering each of these points to more clearly highlight the novelty of our manuscript. We have also added the word “thresholds” to the article title to emphasize the importance of our new results.

(2) As the authors mention, the tuning across different frequencies of afferent responses to transmastoid GVS differs markedly from that to internal GVS. What could be the reason for this difference? A direct comparison of internal vs. external GVS (as performed in Kim and Curthoys, *Brain Res Bull*, 2004) would be required to elucidate this issue in detail. The model of vestibular afferent discharge employed by the authors can apparently not account for this difference. The authors speculate that one reason for this difference could be that vestibular hair cells and afferent fibers are differentially recruited by internal vs. external GVS. What is the rationale for this assumption?

This is indeed an interesting question, but we would like to first emphasize that the Kim and Curthoys study did not address response dynamics (see response to Question 1 above). To directly address the reviewer’s concern, we have revised this section of the text to better motivate our proposal that differential recruitment of vestibular hair cells and afferent fibers might contribute to differences observed for internal versus external GVS. Specifically, we now consider the study in guinea pig by Kim and Curthoys (2004) (see bottom of page 13/top of page 14), who compared the average change in afferent response evoked by constant current steps in both conditions. They found that external GVS is ~10 times less effective than internal stimulation. We thus speculate that we would see an even greater difference in macaque monkey (due to the larger head size) and that such bioelectrical filtering, in part, could also contribute to the disparity between the responses of vestibular afferents to external versus internal GVS.

Next, to more explicitly address the potential mechanisms underlying the dynamics of afferent responses across frequencies, we have also revised the paragraphs in the Discussion that follow (see page 15) to provide a more critical analysis of the observed differences in dynamics. Notably, we now clearly emphasize that while the Smith and Goldberg based model can only partially explain our results, this does not necessarily mean that the responses we see are not predominately produced by activation of the afferent fibers. Indeed, recent studies by Eatock and others have characterized the dynamics of the conductances (e.g., K-LV, HCN, S-K and other K conductances) of vestibular afferent neurons. The dynamics of these ion channels will not be fully captured by the simple Smith and Goldberg model. It is noteworthy that our data showed steeper gain curves (more increase with frequency), as well as a significant phase shift at lower frequencies, than could not be predicted by the Smith and Goldberg model. As discussed by Songer & Eatock 2013, the spike-generating ion channels in afferents could contribute to generating high-pass filtering gain and phase leads. For example, KLV conductances can produce more negative spike thresholds resulting in increased phase lead (Songer & Eatock 2013).

Consistent with this proposal, KLV conductances are more densely expressed by irregular neurons (Kalluri et al., 2010; Lysakowski et al., 2011; Songer and Eatock, 2013), which show greater phase leads.

Further, we note that based on the work from the Straka group in *Xenopus* (Gensberger et al., 2016), it is also likely that GVS activates hair cells (see third paragraph page 15). These investigators estimated that ~30% of the change in afferent firing was the result of direct hair cell activation. The idea that GVS activates hair cells is also consistent with the conclusions of GVS studies in humans with gentamicin-induced hair cell loss (Aw et al., 2008). One possible mechanism by which this could occur is by retrograde GVS activation of nonquantal transmission (Songer and Eatock, 2013; Contini et al., 2017; Eatock, 2018), which is faster than conventional quantal transmission and thus could also potentially contribute to increased phase leads (Songer and Eatock, 2013).

Thus, to address the reviewer's concerns, we have revised the Discussion to include each of these points.

(3) A recent study indicates that GVS does not bypass vestibular hair cells but indeed activates them in a frequency-dependent manner (see Gensberger et al., *J Neurosci*, 2016). How could this piece of information be incorporated into the model simulations employed in the present study to more adequately estimate the response pattern of vestibular end-organs to GVS?

As reviewed above, it is likely that both i) the dynamics of the conductances (e.g., K-LV, HCN, S-K and other K conductances) of vestibular afferent neurons, as well as ii) the direct activation of vestibular hair cells likely contribute to the high pass responses observed as a function of increasing frequency. We now more explicitly discuss these possibilities in the revised Discussion. In our study, we use a relatively simple stochastic model to provide a sense of how we can predict our results using a model that remains a standard in the vestibular field. We were surprised that this simple model can reproduce the general trends we observed as a function of frequency. However, as reviewed in our response above, we also now clearly discuss its limitations and the fact that this simple model does not fully capture the high-pass nature of the afferent responses evoked by GVS. We note that, at this time, it is premature to generate a model at a more reduced level since it would not only need to account for hair cell activation, but it would also need to include numerous conductances, and account for the complexity of the dynamic interactions between them, as well as geometry of their distributions with different afferents. These requirements fall short of our current state of knowledge and thus creating such a model would require making far too many assumptions to be feasible. Nevertheless, we agree that this will be an interesting and important direction for future work and now mention this in our revised Discussion (see third paragraph page 15).

(4) Why did the authors not compare eye movement responses to GVS vs. natural motion stimulation as in previous studies (e.g. Gensberger et al., *J Neurosci*, 2016). This would provide important information whether the frequency-dependent responses of vestibular afferents to GVS vs. natural motion stimulation are preserved at later stages of vestibular information processing.

We appreciate the suggestion made by the reviewer regarding the comparison of eye movements evoked by GVS and whole-body motion. There are, however, several complications that would arise when performing such a comparison. The experimental model used by Gensberger et al. (2016) allowed them to deliver isolated electrical stimulation to horizontal canal afferents only. As a result, they were

able to make a direct comparison of GVS- and motion-evoked eye movements through equivalent rotation movement in the horizontal plane. Our experiments, in contrast, were designed to establish the effects of transmastoid GVS as delivered in humans. This type of stimulation does not target an individual end organ – in contrast to the acute animal model preparation used in the Gensberger et al. (2016) study. Indeed, our findings using transmastoid GVS show that afferents from all canal and otolith end-organs are affected, whereby afferent firing rates increase on the cathode side and decrease on the anode side. This pattern of afferent activity has no physiological motion equivalent (as considered in our Discussion), which makes it impossible to provide a direct comparison of GVS and motion-evoked eye movements in a manner similar to Gensberger et al. (2016). Nevertheless, we have validated our primate-based model by establishing that GVS in alert macaques primarily evokes torsional eye movements that are comparable to those measured when GVS is applied to human (see response to Question 1 above).

Minor comments:

(1) It is not clear from the methods section what eye tracking technique was used to record eye movement responses to GVS. Was it search coil, video-oculography or both?

We realize that the description of the experimental setup was not clear as to the use of either search-coil or video-oculography measurements. The analysis and presentation of torsional data (see Figure 1) relied on video-oculography measurements. Search coil data were used to confirm that eye position responses in the horizontal and vertical directions were similar across measurement platforms. We have made changes to the methods to clarify this issue on page 20 (third paragraph).

(2) What were the exact limitations of the experimental setup that precluded linear motion stimulation along the vertical axis?

The motion platform used for these experiments did not allow for motion of the monkey in the vertical axis. This has been clarified in the text on page 20 (fourth paragraph).

(3) Please correct: “To test eye movement recordings to sinusoidal GVS ...” (line 503)

We have made the appropriate correction to this text.

(4) Please correct: “The motion-GVS ration of both regular and irregular canal afferents, as well as of irregular afferents, decreases as a function of frequency ...” (lines 863-865)

We have made the appropriate correction to this text.

Reviewer #2 (Remarks to the Author):

Galvanic vestibular stimulation (GVS) has been used in many studies in humans as a means of activating the vestibular system without having to physically move the head and/or body. A major limitation of these studies is that it has been almost completely unknown how GVS induces activity in the vestibular system, thus making its effects on behavior difficult to interpret. This study from the Cullen lab applies GVS to monkeys and records the activity of vestibular afferents to both GVS and physical motion stimuli. While this is only a first step in understanding how GVS activates the vestibular system, it is a very important first step. Notably, the results show that GVS similarly activates both canal and otolith

afferents, which is not what had been assumed in many studies. Thus, this study presents a very important advance for understanding how GVS impacts vestibularly-driven behavior.

Overall, I find this to be a well-conducted study that is a valuable contribution to the literature. I have a couple of substantive concerns that should be addressed, as well as a few minor issues.

Major comments:

1) I was a bit disappointed that the authors only characterized neural responses in terms of gain and phase. It would also be quite valuable to know the effect of GVS on neural sensitivity across frequencies. As the Cullen lab has shown previously very nicely, the selectivity profile of response gain and sensitivity (signal/noise ratio) are not necessarily the same. In order to help these findings be interpreted in terms of effects of GVS on behavioral sensitivity, it would also be quite useful for the authors to quantify neuronal sensitivity and present these results also.

We thank the reviewer for this suggestion, and have completed the suggested analysis. We agree that the results of this analysis will be of considerable interest to investigators in the field. In particular, there is a current research focus on understanding the perceptual consequences of GVS, which we now consider in our revised Discussion. The measurement of neural detection thresholds, which are a measure of signal-to-noise, provides a significant advance for the interpretation of these studies.

Specifically, to directly address the reviewers comment, we have performed a complete analysis of each canal and otolith afferent in our data set, following the same approach detailed in our previous work (Jamali et al., 2013; Jamali et al., 2014). Overall these results show that primary vestibular afferent thresholds vary minimally with the frequency of GVS, the firing rate regularity or the end organ the afferents innervate. Importantly, this indicates that changes in sensitivity overcome any adverse effects of increased intrinsic neuronal variability for all afferents, such that all afferents transmit equivalent levels of information to central vestibular pathways for the detection of GVS-evoked sensations of self-motion. We also found that neuronal thresholds were lower than direction recognition thresholds estimated in humans using GVS (Peters et al., 2015). This contrasts with an equivalent comparison of neuronal afferent and human perceptual thresholds during natural translation motion (see Jamali et al. 2013), which for neuronal thresholds are always higher than human perception. These differences between neuronal vs human perceptual thresholds to GVS and natural motion may be related to differences in end organ morphology as well as head size/bone thickness between the monkey and humans. More importantly, it highlights the importance of addressing how central pathways integrate this unnatural vestibular afferent input to form a perception of movement, which will be an important direction for future work. The additional insight provided by these new results has led us to include the word “threshold” in our title to highlight this outcome.

2) I found there to be some tension and inconsistency among the way that the authors frame and state some of their conclusions. In comparing responses to GVS and real motion stimuli (e.g., Figs. 4&5), the authors repeatedly describe the results as “marked differences” between GVS and motion. On the other hand, when comparing measured responses to GVS with the model predictions (Fig. 6), the authors describe these differences as “minor discrepancies”. To me, the differences between data and model in Fig. 6 look just as large as the differences between responses to GVS and motion, so these characterizations seemed rather arbitrary, non-rigorous, and somewhat self-serving. If it is important to make such distinctions, then the authors will need to quantify the deviations between datasets appropriately and make these comparisons quantitatively. I would argue that the differences in

frequency tuning between GVS and real motion stimuli are not as dramatic as the authors seem to want to make them seem; I was actually surprised that they are as similar as they are.

We appreciate the reviewer's concern regarding the perceived inconsistencies; we were initially surprised that our simple Smith and Goldberg based model demonstrated similar frequency-dependent trends and appreciate that we neglected to emphasize the differences in dynamics. To address this issue we have 1) represented the differences between GVS and real motion more objectively by removing the terms "marked" or "markedly" when describing them, and 2) addressed the limitations with the model predictions by highlighting that this simple model does not fully capture the high-pass nature of the afferent responses evoked by GVS. Notably, as described in the answer to reviewer 1's second comment (and as detailed in the manuscript on page 15), the model does not fully capture the dynamics of ion channel conductances (e.g., K-LV, HCN, S-K and other K conductances). This may be important since the spike-generating ion channels in afferents could contribute to generating high-pass filtering gain and phase leads (Songer and Eatock, 2013). For example, KLV conductances can produce more negative spike thresholds resulting in increased phase lead (Songer & Eatock 2013). Consistent with this proposal, KLV conductances are more densely expressed by irregular neurons (Kalluri et al., 2010; Lysakowski et al., 2011; Songer and Eatock, 2013), which show greater phase leads.

Minor comments:

3) P. 7: I found the reporting of the (non-significant) statistics here, e.g., " $p > 0.0056$, Bonferroni's correction...", to be unsatisfactory. This doesn't give the reader any clear sense of how close or far these results are from being significant. This is the reason that there is a push toward reporting exact p values recently.

We agree with the reviewer, and now provide a more detailed description of the outcomes of our statistical analyses. Given the number of tests performed we are hesitant to burden the text with a long list of statistics, and thus have instead opted to summarize the stats for each figure in a supplementary table following the format used previously by Voroslakos M et al. (Nature Communications, 2018). This will allow readers to assess how close or far these results are from significance if they wish. In addition, we have made reference to the table in the methods section on page 26.

4) Lines 341-343: the last sentence of this paragraph is not a proper sentence grammatically

We have changed the sentence identified by the reviewer to:

"These results further emphasize that vestibular afferent dynamic responses to GVS cannot be predicted from previously reported responses to internal stimulation."

5) Line 550: there is no 'gain' variable in Equation 5

We thank the reviewer for noting this omission. In the text that the term 'S' represents the gain in Equation 5. We have modified the text to remedy this issue.

Reviewer #3 (Remarks to the Author):

This paper offers a very useful addition to previous work of Goldberg and colleagues. Rather than applying current invasively within the tissue, here it is applied cutaneously to characterising the transfer function between an externally applied current and the resulting change in vestibular nerve activity. The ultimate rationale is to better understand the effects of GVS upon human vestibular nerve activity, where the technique is limited to cutaneous stimulation. The methods and approach seem sophisticated

and sound. I have a few queries below. In particular, the authors should highlight any limitations/differences in applying these findings to humans, since this is a key point of the paper. For example, the transfer function of GVS may be tuned to different peripheral mechanics in monkeys eg smaller canals etc.

We agree with the reviewer and have added a number of statements throughout the Discussion to highlight potential limitations/differences in applying our findings to human studies. Specifically, in addressing Reviewer 2's comment (see above), we have analyzed the detection thresholds of vestibular afferents responses to GVS. This novel analysis has allowed us to more directly compare our responses to human studies (see pages 16-18). Specifically, the similarity in detection thresholds across all afferent types suggest that despite the lower sensitivity of regular afferents to GVS compared to irregular afferents, they transmit equivalent information to central vestibular pathways for the detection of GVS-evoked sensations of self-motion. In addition, our neuronal thresholds to GVS were 30-60% lower than previous reports of human direction recognition thresholds. This is now highlighted in the revised manuscript: "This contrasts with neuronal thresholds to natural motion, which are typically larger than human perceptual thresholds and require integration of multiple afferents by higher order brain areas to reach equivalent perceptual levels. These differences between neuronal vs human perceptual thresholds to GVS and natural motion may be related to end organ morphology as well as the larger head size and thicker bone of humans." We have also provided additional discussion regarding the similarities in transfer functions of GVS-to-canal and GVS-to-otoliths to propose how we expect GVS to influence the vestibular afferents (see first paragraph page 16). This additional discussion point also addresses comment 4 from the reviewer (please see below). We believe these additions help strengthen the paper and provide important insight to potential readers.

- Line 108-9: You mentioning that you mainly saw torsional eye movement, and that this agrees with prior studies. However, this is slightly disingenuous because, as you state in the figure legend/methods, the monkeys were fixating on a light which would suppress any vertical/lateral movement. For transparency, I suggest you mention the fixation light here in the results text too.

We agree with the reviewer and have added a clarifying statement in the results section regarding the use of a visual target during these trials (see page 5, second paragraph).

- Figure 1C (inset) & Discussion: The authors point out that the absolute gain is quite a bit higher for Monkey B. They speculate anatomical differences, but don't say whether they looked for this in the monkey e.g. Did this monkey have a smaller head/thinner skull?

The reviewer makes a good point, which we have now explicitly considered (see page 17). However, the size of both monkeys was comparable (4.9 vs. 5.5 Kg for monkeys H and B respectively). Thus, Monkey B was actually the larger of the two animals. Since this difference was consistent across recording sessions, we hypothesize that the difference may arise from the thickness of the skull and/or overlying tissue. Unfortunately, these animals are no longer available to us, so it is not possible to confirm this hypothesis.

- Figure 4 shows that neural response gain increases with frequency for both GVS and real motion. This is perhaps surprising – would you not expect the two stimuli to show opposing effects? i.e. the GVS-nerve transfer function might be expected to compensate for the mechanics of the canals/otoliths? Maybe my thinking is overly simplistic, but nevertheless this could do with some discussion.

We thank the reviewer for this comment and suggestion that GVS and motion stimuli would have opposing effects on afferent responses. The similar transfer functions from GVS-to-canal and GVS-to-otolith afferents despite the different mechanics of the canals/otoliths, however, argues against this possibility. We have modified the Discussion (see page 16) to address this potentially surprising finding. We propose that the GVS-afferent transfer functions are dictated mainly by the properties of the afferents and their responses to the GVS-induced nearby voltage. As the stochastic model of repetitive activity in vestibular afferent did not predict all the dynamics of the observed afferent responses, we argue that other factors including vestibular hair cell recruitment by GVS or the various conductances within the afferents may contribute to the responses. In support of our proposal, we also found that neuronal detection thresholds of vestibular afferents do not vary substantially with discharge regularity or the class of sensory end organ they innervate. This differs from canal afferent thresholds during natural motion, which vary with discharge regularity (Sadeghi et al., 2007; Massot et al., 2011).

- Legend applying to figure 4b: Here it states values of 0.3 vs 0.1 spk/s/deg/s at 1Hz. However, the inset graph values in 4b do not seem to tally with these values, where the minimum value seems to be roughly 0.5 spk/s/deg/s. Please clarify.

The reviewer is correct that these were incorrectly stated in the legend. We have modified the text accordingly.

- Legend figure 7: line 865 – Do you mean ‘decreases’ rather than ‘increases’?

We have made the appropriate correction to this text.

- Figure 6A: I might have expected tissue capacitance to produce changes in gain and phase with frequency, so the flat gain/phase plots of the Head Impedance Dynamics are a little surprising. Can you speculate whether the much larger head of a human being might introduce such capacitive filtering effects, not seen in the monkey?

We agree with the reviewer; when we ran this experiment we also expected to see some evidence of capacitive filtering effects. Results similar to ours, however, have been observed in a recent study in human cadavers during transcranial direct current stimulation. Voroslakos et al. (2018) demonstrate primarily ohmic properties of the brain, surrounding skull and soft tissue with negligible capacitive components (Voroslakos et al., 2018). We believe that the similarity in conductance behaviors of human cadavers and macaque monkeys further validates our primate model for transmastoid GVS. A short statement has been added in the Results (page 10) to highlight these similarities.

Aw ST, Todd MJ, Aw GE, Weber KP, Halmagyi GM (2008) Gentamicin vestibulotoxicity impairs human electrically evoked vestibulo-ocular reflex. *Neurology* 71:1776-1782.

Contini D, Price SD, Art JJ (2017) Accumulation of K(+) in the synaptic cleft modulates activity by influencing both vestibular hair cell and calyx afferent in the turtle. *J Physiol* 595:777-803.

Eatock RA (2018) Specializations for Fast Signaling in the Amniote Vestibular Inner Ear. *Integr Comp Biol* 58:341-350.

Gensberger KD, Kaufmann AK, Dietrich H, Branoner F, Banchi R, Chagnaud BP, Straka H (2016) Galvanic Vestibular Stimulation: Cellular Substrates and Response Patterns of Neurons in the Vestibulo-Ocular Network. *J Neurosci* 36:9097-9110.

- Jamali M, Carriot J, Cullen KE (2013) Impacts of neuronal sensitivity and variability on the encoding of linear self-motion. Society for Neuroscience Abstract Viewer and Itinerary Planner 43.
- Jamali M, Mitchell DE, Dale A, Carriot J, Sadeghi SG, Cullen KE (2014) Neuronal detection thresholds during vestibular compensation: contributions of response variability and sensory substitution. *J Physiol* 592:1565-1580.
- Kalluri R, Xue J, Eatock RA (2010) Ion channels set spike timing regularity of mammalian vestibular afferent neurons. *J Neurophysiol* 104:2034-2051.
- Kim J, Curthoys IS (2004) Responses of primary vestibular neurons to galvanic vestibular stimulation (GVS) in the anaesthetised guinea pig. *Brain Res Bull* 64:265-271.
- Lysakowski A, Gaboyard-Niay S, Calin-Jageman I, Chatlani S, Price SD, Eatock RA (2011) Molecular microdomains in a sensory terminal, the vestibular calyx ending. *J Neurosci* 31:10101-10114.
- Massot C, Chacron MJ, Cullen KE (2011) Information transmission and detection thresholds in the vestibular nuclei: single neurons vs. population encoding. *J Neurophysiol* 105:1798-1814.
- Peters RM, Rasman BG, Inglis JT, Blouin JS (2015) Gain and phase of perceived virtual rotation evoked by electrical vestibular stimuli. *J Neurophysiol* 114:264-273.
- Sadeghi SG, Chacron MJ, Taylor MC, Cullen KE (2007) Neural variability, detection thresholds, and information transmission in the vestibular system. *J Neurosci* 27:771-781.
- Songer JE, Eatock RA (2013) Tuning and timing in mammalian type I hair cells and calyceal synapses. *J Neurosci* 33:3706-3724.
- Voroslakos M, Takeuchi Y, Brinyiczki K, Zombori T, Oliva A, Fernandez-Ruiz A, Kozak G, Kincses ZT, Ivanyi B, Buzsaki G, Berenyi A (2018) Direct effects of transcranial electric stimulation on brain circuits in rats and humans. *Nature communications* 9:483.

Reviewers' Comments:

Reviewer #1:

Remarks to the Author:

Overall this is an improved version of the manuscript and the authors have sufficiently addressed each of the concerns raised during the initial review.

Reviewer #2:

Remarks to the Author:

The authors have been quite responsive to the previous reviews. The addition of the neuronal thresholds to the manuscript makes it stronger. I am satisfied with the authors' responses to my previous comments and have no additional concerns at this time. I think this study makes a valuable contribution to our understanding of the neural effects of GVS.

Reviewer #3:

Remarks to the Author:

I am happy that the authors have addressed my concerns.